# Uncertainty-Based Offline Reinforcement Learning with Diversified Q-Ensemble

**Gaon An**[*][1][2], **Seungyong Moon**[*][1][2], **Jang-Hyun Kim**[1][2], **Hyun Oh Song**[†][1][2][3]

Seoul National University[1]
Neural Processing Research Center[2]
DeepMetrics[3]
{white0234,symoon11,janghyun,hyunoh}@mllab.snu.ac.kr

## Abstract

Offline reinforcement learning (offline RL), which aims to find an optimal policy from a previously collected static dataset, bears algorithmic difficulties due to function approximation errors from out-of-distribution (OOD) data points. To this end, offline RL algorithms adopt either a constraint or a penalty term that explicitly guides the policy to stay close to the given dataset. However, prior methods typically require accurate estimation of the behavior policy or sampling from OOD data points, which themselves can be a non-trivial problem. Moreover, these methods under-utilize the generalization ability of deep neural networks and often fall into suboptimal solutions too close to the given dataset. In this work, we propose an uncertainty-based offline RL method that takes into account the confidence of the Q-value prediction and does not require any estimation or sampling of the data distribution. We show that the clipped Q-learning, a technique widely used in online RL, can be leveraged to successfully penalize OOD data points with high prediction uncertainties. Surprisingly, we find that it is possible to substantially outperform existing offline RL methods on various tasks by simply increasing the number of Q-networks along with the clipped Q-learning. Based on this observation, we propose an ensemble-diversified actor-critic algorithm that reduces the number of required ensemble networks down to a tenth compared to the naive ensemble while achieving state-of-the-art performance on most of the D4RL benchmarks considered.

## 1 Introduction

Over the recent years, deep reinforcement learning (deep RL) has achieved considerable success in various domains such as robotics [20], recommendation systems [6], and strategy games [26]. However, a major drawback of RL algorithms is that they adopt an active learning procedure, where training steps require active interactions with the environment. This trial-and-error procedure can be prohibitive when scaling RL to real-world applications such as autonomous driving and healthcare, as exploratory actions can cause critical damage to the agent or the environment [19]. *Offline* RL, also known as *batch* RL, aims to overcome this problem by learning policies using only previously collected data without further interactions with the environment [2, 11, 19].

Even though offline RL is a promising direction to lead a more *data-driven* way of solving RL problems, recent works show offline RL faces new algorithmic challenges [19]. Typically, if the coverage of the dataset is not sufficient, vanilla RL algorithms suffer severely from extrapolation

---

[*]First two authors have equal contributions

[†]Corresponding author

35th Conference on Neural Information Processing Systems (NeurIPS 2021).

error, overestimating the Q-values of out-of-distribution (OOD) state-action pairs [15]. To this end, most offline RL methods apply some constraints or penalty terms on top of the existing RL algorithms to enforce the learning process to be more conservative. For example, some prior works explicitly regularize the policy to be close to the behavior policy that was used to collect the data [11, 15]. A more recent work instead penalizes the Q-values of OOD state-action pairs to enforce the Q-values to be more pessimistic [16].

While these methods achieve significant performance gains over vanilla RL methods, they either require an estimation of the behavior policy or explicit sampling from OOD data points, which themselves can be non-trivial to solve. Furthermore, these methods do not utilize the generalization ability of the Q-function networks and prohibit the agent from approaching any OOD state-actions without any consideration on whether they are good or bad. However, if we can identify OOD data points where we can predict their Q-values with high confidence, it is more effective not to restrain the agent from choosing those data points.

From this intuition, we propose an uncertainty-based model-free offline RL method that effectively quantifies the uncertainty of the Q-value estimates by an ensemble of Q-function networks and does not require any estimation or sampling of the data distribution. To achieve this, we first show that a well-known technique from online RL, the clipped Q-learning [10], can be successfully leveraged as an uncertainty-based penalization term. Our experiments reveal that we can achieve state-of-the-art performance on various offline RL tasks by solely using this technique with increased ensemble size. To further improve the practical usability of the method, we develop an ensemble diversifying objective that significantly reduces the number of required ensemble networks. We evaluate our proposed method on D4RL benchmarks [9] and verify that the proposed method outperforms the previous state-of-the-art by a large margin on various types of environments and datasets.

## 2  Preliminaries

We consider an environment formulated as a Markov Decision Process (MDP) defined by a tuple $(\mathcal{S}, \mathcal{A}, T, r, d_0, \gamma)$, where $\mathcal{S}$ is the state space, $\mathcal{A}$ is the action space, $T(\mathbf{s}' \mid \mathbf{s}, \mathbf{a})$ is the transition probability distribution, $r : \mathcal{S} \times \mathcal{A} \to \mathbb{R}$ is the reward function, $d_0$ is the initial state distribution, and $\gamma \in (0, 1]$ is the discount factor. The goal of reinforcement learning is to find an optimal policy $\pi(\mathbf{a} \mid \mathbf{s})$ that maximizes the cumulative discounted reward $\mathbb{E}_{\mathbf{s}_t, \mathbf{a}_t} [\sum_{t=0}^{\infty} \gamma^t r(\mathbf{s}_t, \mathbf{a}_t)]$, where $\mathbf{s}_0 \sim d_0(\cdot)$, $\mathbf{a}_t \sim \pi(\cdot \mid \mathbf{s}_t)$, and $\mathbf{s}_{t+1} \sim T(\cdot \mid \mathbf{s}_t, \mathbf{a}_t)$.

One of the major approaches for obtaining such a policy is Q-learning [12, 20] which learns a state-action value function $Q_\phi(\mathbf{s}, \mathbf{a})$ parameterized by a neural network that represents the expected cumulative discounted reward when starting from state $\mathbf{s}$ and action $\mathbf{a}$. Standard actor-critic approach [14] learns this Q-function by minimizing the Bellman residual $(Q_\phi(\mathbf{s}, \mathbf{a}) - \mathcal{B}^{\pi_\theta} Q_\phi(\mathbf{s}, \mathbf{a}))^2$, where $\mathcal{B}^{\pi_\theta} Q_\phi(\mathbf{s}, \mathbf{a}) = \mathbb{E}_{\mathbf{s}' \sim T(\cdot \mid \mathbf{s}, \mathbf{a})} [r(\mathbf{s}, \mathbf{a}) + \gamma \mathbb{E}_{\mathbf{a}' \sim \pi_\theta(\cdot \mid \mathbf{s}')} Q_\phi(\mathbf{s}', \mathbf{a}')]$ is the Bellman operator. In the context of offline RL, where transitions are sampled from a static dataset $\mathcal{D}$, the objective for the Q-network becomes minimizing

$$J_q(Q_\phi) := \mathbb{E}_{(\mathbf{s}, \mathbf{a}, \mathbf{s}') \sim \mathcal{D}} \left[ \left( Q_\phi(\mathbf{s}, \mathbf{a}) - \left( r(\mathbf{s}, \mathbf{a}) + \gamma \mathbb{E}_{\mathbf{a}' \sim \pi_\theta(\cdot \mid \mathbf{s}')} [Q_{\phi'}(\mathbf{s}', \mathbf{a}')] \right) \right)^2 \right], \qquad (1)$$

where $Q_{\phi'}$ represents the target Q-network softly updated for algorithmic stability [20]. The policy, which is also parameterized by a neural network, is updated in an alternating fashion to maximize the expected Q-value: $J_p(\pi_\theta) := \mathbb{E}_{\mathbf{s} \sim \mathcal{D}, \mathbf{a} \sim \pi_\theta(\cdot \mid \mathbf{s})} [Q_\phi(\mathbf{s}, \mathbf{a})]$.

However, as the policy is updated to maximize the Q-values, the actions $\mathbf{a}'$ sampled from the current policy in Equation (1) can be biased towards OOD actions with erroneously high Q-values. In the offline RL setting, such errors cannot be corrected by feedback from the environment as in online RL. To handle the error propagation from these OOD actions, most offline RL algorithms regularize either the policy [11, 15] or the Q-function [16] to be biased towards the given dataset. However, the policy regularization methods typically require an accurate estimation of the behavior policy. The previous state-of-the-art method CQL [16] instead learns conservative Q-values without estimating the behavior policy by penalizing the Q-values of OOD actions by

$$\min_\phi \ J_q(Q_\phi) + \alpha \Big( \mathbb{E}_{\mathbf{s} \sim \mathcal{D}, \mathbf{a} \sim \mu(\cdot \mid \mathbf{s})} [Q_\phi(\mathbf{s}, \mathbf{a})] - \mathbb{E}_{(\mathbf{s}, \mathbf{a}) \sim \mathcal{D}} [Q_\phi(\mathbf{s}, \mathbf{a})] \Big),$$

where $\mu$ is an approximation of the policy that maximizes the current Q-function. While CQL does not need explicit behavior policy estimation, it requires sampling from an appropriate action distribution $\mu(\cdot \mid \mathbf{s})$.

# 3 Uncertainty penalization with Q-ensemble

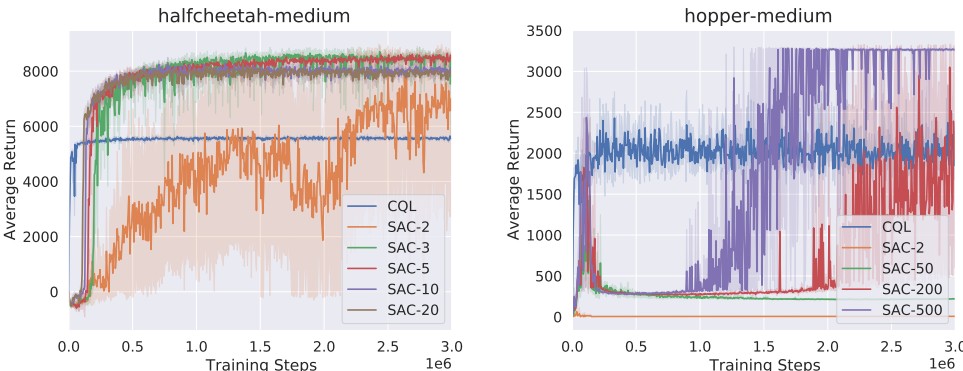

Figure 1: Performance of SAC-$N$ on halfcheetah-medium and hopper-medium datasets while varying $N$, compared to CQL. 'Average Return' denotes the undiscounted return of each policies on evaluation. Results averaged over 4 seeds.

In this section, we turn our attention to a conventional technique from online RL, Clipped Double Q-Learning [10], which uses the minimum value of two parallel Q-networks as the Bellman target: $y = r(\mathbf{s}, \mathbf{a}) + \gamma \, \mathbb{E}_{\mathbf{a}' \sim \pi_\theta(\cdot \mid \mathbf{s}')} \left[ \min_{j=1,2} Q_{\phi'_j}(\mathbf{s}', \mathbf{a}') \right]$. Although this technique was originally proposed in online RL to mitigate the overestimation from general prediction errors, some offline RL algorithms [11, 15, 28] also utilize this technique to enforce their Q-value estimates to be more pessimistic. However, the isolated effect of the clipped Q-learning in offline RL was not fully analyzed in the previous works, as they use the technique only as an auxiliary term that adds up to their core methods.

To examine the ability of clipped Q-learning to prevent the overestimation in offline RL on its own, we modify SAC [12] by increasing the number of Q-ensembles from 2 to $N$:

$$\min_{\phi_i} \mathbb{E}_{\mathbf{s},\mathbf{a},\mathbf{s}' \sim \mathcal{D}} \left[ \left( Q_{\phi_i}(\mathbf{s}, \mathbf{a}) - \left( r(\mathbf{s}, \mathbf{a}) + \gamma \, \mathbb{E}_{\mathbf{a}' \sim \pi_\theta(\cdot \mid \mathbf{s}')} \left[ \min_{j=1,\dots,N} Q_{\phi'_j}(\mathbf{s}', \mathbf{a}') - \beta \log \pi_\theta(\mathbf{a}' \mid \mathbf{s}') \right] \right) \right)^2 \right]$$

$$\max_\theta \mathbb{E}_{\mathbf{s} \sim \mathcal{D}, \mathbf{a} \sim \pi_\theta(\cdot \mid \mathbf{s})} \left[ \min_{j=1,\dots,N} Q_{\phi_j}(\mathbf{s}, \mathbf{a}) - \beta \log \pi_\theta(\mathbf{a} \mid \mathbf{s}) \right], \tag{2}$$

for $i = 1, \dots, N$. We denote this modified algorithm as SAC-$N$.

Figure 1 shows the preliminary experiments on D4RL halfcheetah-medium and hopper-medium datasets [9] while varying $N$. Note that these datasets are constructed from suboptimal behavior policies. Surprisingly, as we gradually increase $N$, we can successfully find policies that outperform the previous state-of-the-art method (CQL) by a large margin. In fact, as we will present in Section 5, SAC-$N$ outperforms CQL on various types of environments and data-collection policies.

To understand why this simple technique works so well, we can first interpret the clipping procedure (choosing the minimum value from the ensemble) as penalizing state-action pairs with high-variance Q-value estimates, which encourages the policy to favor actions that appeared in the dataset [11]. The dataset samples will naturally have lower variance compared to the OOD samples as the Bellman residual term in Equation (2) explicitly aligns the Q-value predictions for the dataset samples. More formally, we can regard this difference in variance as accounting for *epistemic uncertainty* [8] which refers to the uncertainty stemming from limited data and knowledge.

Utilization of the clipped Q-value relates to methods that consider the confidence bound of the Q-value estimates [24]. Online RL methods typically utilize the Q-ensemble to form an optimistic estimate of the Q-value, by adding the standard deviation to the mean of the Q-ensembles [18]. This optimistic Q-value, also known as the upper-confidence bound (UCB), can encourage the exploration

of unseen actions with high uncertainty. However, in offline RL, the dataset available during training is fixed, and we have to focus on *exploiting* the given data. For this purpose, it is natural to utilize the lower-confidence bound (LCB) of the Q-value estimates, for example by subtracting the standard deviation from the mean, which allows us to avoid risky state-actions.

The clipped Q-learning algorithm, which chooses the worst-case Q-value instead to compute the pessimistic estimate, can also be interpreted as utilizing the LCB of the Q-value predictions. Suppose $Q(\mathbf{s}, \mathbf{a})$ follows a Gaussian distribution with mean $m(\mathbf{s}, \mathbf{a})$ and standard deviation $\sigma(\mathbf{s}, \mathbf{a})$. Also, let $\{Q_j(\mathbf{s}, \mathbf{a})\}_{j=1}^{N}$ be realizations of $Q(\mathbf{s}, \mathbf{a})$. Then, we can approximate the expected minimum of the realizations following the work of Royston [23] as

$$\mathbb{E}\left[\min_{j=1,\dots,N} Q_j(\mathbf{s}, \mathbf{a})\right] \approx m(\mathbf{s}, \mathbf{a}) - \Phi^{-1}\left(\frac{N - \frac{\pi}{8}}{N - \frac{\pi}{4} + 1}\right)\sigma(\mathbf{s}, \mathbf{a}), \tag{3}$$

where $\Phi$ is the CDF of the standard Gaussian distribution. This relation indicates that using the clipped Q-value is similar to penalizing the ensemble mean of the Q-values with the standard deviation scaled by a coefficient dependent on $N$.

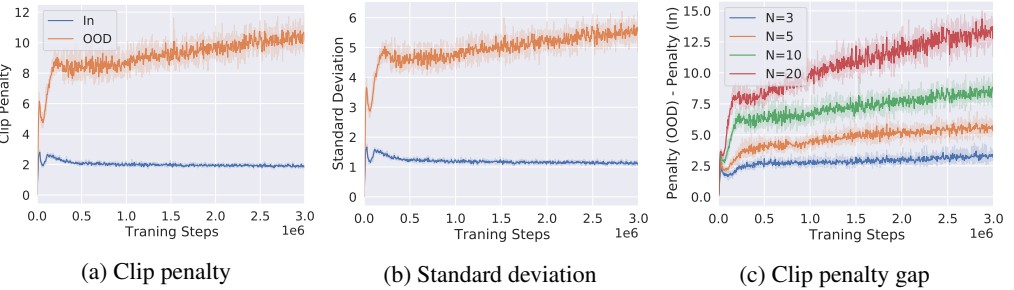

(a) Clip penalty  (b) Standard deviation  (c) Clip penalty gap

Figure 2: (a) and (b) each plots the size of the clip penalty and the standard deviation of the Q-value estimates for in-distribution (behavior) and OOD (random) actions while training SAC-10 on halfcheetah-medium dataset. (c) plots the gap of the clip penalty between the in-distribution and OOD actions while varying $N$. Results averaged over 4 seeds.

We now move on to the empirical analysis of the clipped Q-learning. Figure 2a compares the strength of the uncertainty penalty on in-distribution and OOD actions. Specifically, we compare actions sampled from two types of policies: (1) the behavior policy which was used to collect the dataset, and (2) the random policy which samples actions uniformly from the action space. For each policy, we measure the size of the penalty from the clipping as $\mathbb{E}_{\mathbf{s}\sim\mathcal{D}, \mathbf{a}\sim\pi(\cdot|\mathbf{s})}\left[\frac{1}{N}\sum_{j=1}^{N} Q_{\phi_j}(\mathbf{s}, \mathbf{a}) - \min_{j=1,\dots,N} Q_{\phi_j}(\mathbf{s}, \mathbf{a})\right]$. Figure 2a shows that the clipping term penalizes the random state-action pairs much stronger than the in-distribution pairs throughout the training. For comparison, we also measure the standard deviation of the Q-values for each policy. The results in Figure 2b show that as we conjectured, the Q-value predictions for the OOD actions have a higher variance. We also find that the size of the penalty and the standard deviation are highly correlated, as we noted in Equation (3).

As we observe that OOD actions have higher variance on Q-value estimates, the effect of increasing $N$ becomes obvious: it strengthens the penalty applied to the OOD samples compared to the dataset samples. To verify this, we measured the relative penalty applied to the OOD samples in Figure 2c and found that indeed the OOD samples are penalized relatively further as $N$ increases.

## 4 Ensemble gradient diversification

Even though SAC-$N$ outperforms existing methods on various tasks, it sometimes requires an excessively large number of ensembles to learn stably (*e.g.*, $N = 500$ for hopper-medium). While investigating its reason, we found that the performance of SAC-$N$ is negatively correlated with the degree to which the input gradients of Q-functions $\nabla_{\mathbf{a}} Q_{\phi_j}(\mathbf{s}, \mathbf{a})$ are aligned, which decreases with $N$. Figure 4 measures the minimum cosine similarity between the gradients of the Q-functions $\min_{i \neq j}\langle\nabla_{\mathbf{a}} Q_{\phi_i}(\mathbf{s}, \mathbf{a}), \nabla_{\mathbf{a}} Q_{\phi_j}(\mathbf{s}, \mathbf{a})\rangle$ to examine the alignment of the gradients while varying $N$ on the D4RL hopper-medium dataset. The results imply that the performance of the learned policy degrades significantly when the Q-functions share a similar local structure.

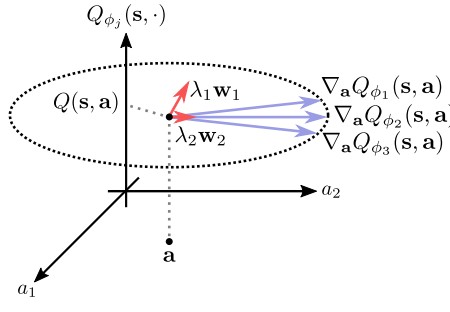

Var$(Q_{\phi_j}(\mathbf{s}, \mathbf{a} + k\mathbf{w_2}))$ is small so that $\mathbf{a} + k\mathbf{w_2}$ is not sufficiently penalized.

(a) Without ensemble gradient diversification

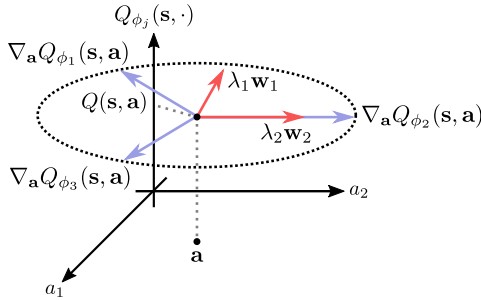

Var$(Q_{\phi_j}(\mathbf{s}, \mathbf{a} + k\mathbf{w}))$ is large for every direction $\mathbf{w}$ so that all OOD actions are sufficiently penalized.

(b) With ensemble gradient diversification

Figure 3: Illustration of the ensemble gradient diversification. The vector $\lambda_i \mathbf{w}_i$ represents the normalized eigenvector $\mathbf{w}_i$ of Var$(\nabla_\mathbf{a} Q_{\phi_j}(\mathbf{s}, \mathbf{a}))$ multiplied by its eigenvalue $\lambda_i$.

We now show that the alignment of the input gradients can induce insufficient penalization of near-distribution data points, which leads to requiring a large number of ensemble networks. Let $\nabla_\mathbf{a} Q_{\phi_j}(\mathbf{s}, \mathbf{a})$ be the gradient of the $j$-th Q-function with respect to the behavior action $\mathbf{a}$ and assume the gradient is normalized for simplicity. If the gradients of the Q-functions are well-aligned as illustrated in Figure 3a, then there exists a unit vector $\mathbf{w}$ such that the Q-values for the OOD actions along the direction of $\mathbf{w}$ have a low variance. To show this, we first assume the Q-value predictions for the in-distribution state-action pairs coincide, *i.e.*, $Q_{\phi_j}(\mathbf{s}, \mathbf{a}) = Q(\mathbf{s}, \mathbf{a})$ for $j = 1, \ldots, N$. Note that this can be optimized by minimizing the Bellman error. Then, using the first-order Taylor approximation, the sample variance of the Q-values at an OOD action along $\mathbf{w}$ can be represented as

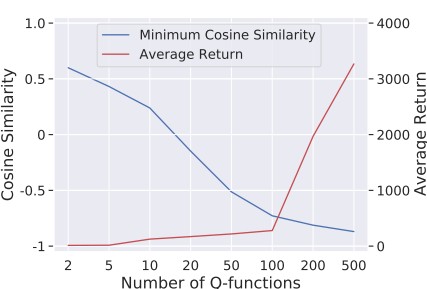

Figure 4: Plot of the minimum cosine similarity between the input gradients of Q-functions and the average return while varying the number of Q-functions.

$$\text{Var}\left(Q_{\phi_j}(\mathbf{s}, \mathbf{a} + k\mathbf{w})\right) \approx \text{Var}\left(Q_{\phi_j}(\mathbf{s}, \mathbf{a}) + k\left\langle \mathbf{w}, \nabla_\mathbf{a} Q_{\phi_j}(\mathbf{s}, \mathbf{a})\right\rangle\right)$$
$$= \text{Var}\left(Q(\mathbf{s}, \mathbf{a}) + k\left\langle \mathbf{w}, \nabla_\mathbf{a} Q_{\phi_j}(\mathbf{s}, \mathbf{a})\right\rangle\right)$$
$$= k^2 \text{Var}\left(\left\langle \mathbf{w}, \nabla_\mathbf{a} Q_{\phi_j}(\mathbf{s}, \mathbf{a})\right\rangle\right)$$
$$= k^2 \mathbf{w}^\mathsf{T} \text{Var}\left(\nabla_\mathbf{a} Q_{\phi_j}(\mathbf{s}, \mathbf{a})\right) \mathbf{w},$$

where $\langle \cdot, \cdot \rangle$ denotes an inner-product, $k \in \mathbb{R}$, and Var$\left(\nabla_\mathbf{a} Q_{\phi_j}(\mathbf{s}, \mathbf{a})\right)$ is the sample variance matrix for the input gradients $\nabla_\mathbf{a} Q_{\phi_j}(\mathbf{s}, \mathbf{a})$. One interesting property of the variance matrix is that its total variance, which is equivalent to the sum of its eigenvalues, can be represented as a function of the norm of the average gradients by Lemma 1.

**Lemma 1.** *The total variance of the matrix* Var$\left(\nabla_\mathbf{a} Q_{\phi_j}(\mathbf{s}, \mathbf{a})\right)$ *is equal to* $1 - \|\bar{q}\|_2^2$, *where* $\bar{q} = \frac{1}{N} \sum_{j=1}^N \nabla_\mathbf{a} Q_{\phi_j}(\mathbf{s}, \mathbf{a})$.

Let $\lambda_{\min}$ be the smallest eigenvalue of Var$\left(\nabla_\mathbf{a} Q_{\phi_j}(\mathbf{s}, \mathbf{a})\right)$ and $\mathbf{w}_{\min}$ be the corresponding normalized eigenvector. Also, let $\epsilon > 0$ be the value such that $\min_{i \neq j} \left\langle \nabla_\mathbf{a} Q_{\phi_i}(\mathbf{s}, \mathbf{a}), \nabla_\mathbf{a} Q_{\phi_j}(\mathbf{s}, \mathbf{a})\right\rangle = 1 - \epsilon$. Then, using Lemma 1, we can prove that the variance of the Q-values for an OOD action along $\mathbf{w}_{\min}$ is upper-bounded by some constant multiple of $\epsilon$, which is given by Proposition 1.

**Proposition 1.** *Suppose* $Q_{\phi_j}(\mathbf{s}, \mathbf{a}) = Q(\mathbf{s}, \mathbf{a})$ *and* $Q_{\phi_j}(\mathbf{s}, \cdot)$ *is locally linear in the neighborhood of* $\mathbf{a}$ *for all* $j \in [N]$. *Let* $\lambda_{\min}$ *and* $\mathbf{w}_{\min}$ *be the smallest eigenvalue and the corresponding normalized eigenvector of the matrix* Var$\left(\nabla_\mathbf{a} Q_{\phi_j}(\mathbf{s}, \mathbf{a})\right)$ *and* $\epsilon > 0$ *be the value such that*

$\min_{i \neq j} \langle \nabla_{\mathbf{a}} Q_{\phi_i}(\mathbf{s}, \mathbf{a}), \nabla_{\mathbf{a}} Q_{\phi_j}(\mathbf{s}, \mathbf{a}) \rangle = 1 - \epsilon$. *Then, the variance of the Q-values for an OOD action in the neighborhood along the direction of* $\mathbf{w}_{\min}$ *is upper-bounded as follows:*

$$\text{Var}\left( Q_{\phi_j}(\mathbf{s}, \mathbf{a} + k\mathbf{w}_{\min}) \right) \leq \frac{1}{|\mathcal{A}|} \frac{N-1}{N} k^2 \epsilon,$$

*where* $|\mathcal{A}|$ *is the action space dimension.*

We provide the proofs in Appendix A.1. Proposition 1 implies that if there exists such $\epsilon > 0$ that is small, which means the gradients of Q-function are well-aligned, then the variance of the Q-values for an OOD action along a specific direction will also be small. This in turn degrades the ability of the ensembles to penalize OOD actions, which ultimately leads to requiring a large number of ensemble networks.

To address this problem, we propose a regularizer that effectively increases the variance of the Q-values for near-distribution OOD actions. Note that the variance is lower-bounded by some constant multiple of the smallest eigenvalue $\lambda_{\min}$:

$$\begin{aligned}
\text{Var}\left( Q_{\phi_j}(\mathbf{s}, \mathbf{a} + k\mathbf{w}) \right) &\approx k^2 \mathbf{w}^{\mathsf{T}} \text{Var}\left( \nabla_{\mathbf{a}} Q_{\phi_j}(\mathbf{s}, \mathbf{a}) \right) \mathbf{w} \\
&\geq k^2 \mathbf{w}_{\min}^{\mathsf{T}} \text{Var}\left( \nabla_{\mathbf{a}} Q_{\phi_j}(\mathbf{s}, \mathbf{a}) \right) \mathbf{w}_{\min} \\
&= k^2 \lambda_{\min}.
\end{aligned}$$

Therefore, an obvious way to increase this variance is to maximize the smallest eigenvalue of $\text{Var}\left( \nabla_{\mathbf{a}} Q_{\phi_j}(\mathbf{s}, \mathbf{a}) \right)$, which can be formulated as

$$\underset{\phi}{\text{maximize}} \; \mathbb{E}_{\mathbf{s}, \mathbf{a} \sim \mathcal{D}} \left[ \lambda_{\min}\left( \text{Var}\left( \nabla_{\mathbf{a}} Q_{\phi_j}(\mathbf{s}, \mathbf{a}) \right) \right) \right],$$

where $\phi$ denotes the collection of the parameters $\{\phi_j\}_{j=1}^N$. There are several methods to compute the smallest eigenvalue, such as the power method or the QR algorithm [27]. However, these iterative methods require constructing huge computation graphs, which makes optimizing the eigenvalue using back-propagation inefficient. Instead, we aim to maximize the sum of all eigenvalues, which is equal to the total variance. By Lemma 1, it is equivalent to minimizing the norm of the average gradients:

$$\underset{\phi}{\text{minimize}} \; \mathbb{E}_{\mathbf{s}, \mathbf{a} \sim \mathcal{D}} \left[ \left\langle \frac{1}{N} \sum_{i=1}^N \nabla_{\mathbf{a}} Q_{\phi_i}(\mathbf{s}, \mathbf{a}), \frac{1}{N} \sum_{j=1}^N \nabla_{\mathbf{a}} Q_{\phi_j}(\mathbf{s}, \mathbf{a}) \right\rangle \right]. \tag{4}$$

With simple modification, we can reformulate Equation (4) as diversifying the gradients of each Q-function network for in-distribution actions:

$$\underset{\phi}{\text{minimize}} \; J_{\text{ES}}(Q_\phi) := \mathbb{E}_{\mathbf{s}, \mathbf{a} \sim \mathcal{D}} \left[ \frac{1}{N-1} \sum_{1 \leq i \neq j \leq N} \underbrace{\left\langle \nabla_{\mathbf{a}} Q_{\phi_i}(\mathbf{s}, \mathbf{a}), \nabla_{\mathbf{a}} Q_{\phi_j}(\mathbf{s}, \mathbf{a}) \right\rangle}_{\text{ES}_{\phi_i, \phi_j}(\mathbf{s}, \mathbf{a})} \right].$$

Concretely, our final objective can be interpreted as measuring the pairwise alignment of the gradients using cosine similarity, which we denote as the Ensemble Similarity (ES) metric $\text{ES}_{\phi_i, \phi_j}(\mathbf{s}, \mathbf{a})$, and minimizing the ES values for every pair in the Q-ensemble with regard to the dataset state-actions. The illustration of the ensemble gradient diversification is shown in Figure 3b. Note that we instead maximize the total variance to reduce the computational burden. Nevertheless, the modified objective is closely related to maximizing the smallest eigenvalue. The detailed explanation can be found in Appendix A.2.

We name the resulting actor-critic algorithm as Ensemble-Diversified Actor Critic (EDAC) and present the detailed procedure in Algorithm 1 (differences with the original SAC algorithm marked in blue). Note that Algorithm 1 reduces to SAC-$N$ when $\eta = 0$, and further reduces to vanilla SAC when also $N = 2$.

## 5 Experiments

We evaluate our proposed methods against the previous offline RL algorithms on the standard D4RL benchmark [9] . Concretely, we perform our evaluation on MuJoCo Gym (Section 5.1) and Adroit

---

**Algorithm 1** Ensemble-Diversified Actor Critic (EDAC)

---

1: Initialize policy parameters $\theta$, Q-function parameters $\{\phi_j\}_{j=1}^N$, target Q-function parameters $\{\phi_j'\}_{j=1}^N$, and offline data replay buffer $\mathcal{D}$
2: **repeat**
3:     Sample a mini-batch $B = \{(\mathbf{s}, \mathbf{a}, r, \mathbf{s}')\}$ from $\mathcal{D}$
4:     Compute target Q-values (shared by all Q-functions):

$$y(r, \mathbf{s}') = r + \gamma \left( \min_{j=1,\ldots,N} Q_{\phi_j'}\left(\mathbf{s}', \mathbf{a}'\right) - \beta \log \pi_\theta\left(\mathbf{a}' \mid \mathbf{s}'\right) \right), \quad \mathbf{a}' \sim \pi_\theta\left(\cdot \mid \mathbf{s}'\right)$$

5:     Update each Q-function $Q_{\phi_i}$ with gradient descent using

$$\nabla_{\phi_i} \frac{1}{|B|} \sum_{(\mathbf{s}, \mathbf{a}, r, \mathbf{s}') \in B} \left( \left(Q_{\phi_i}\left(\mathbf{s}, \mathbf{a}\right) - y\left(r, \mathbf{s}'\right)\right)^2 + \frac{\eta}{N-1} \sum_{1 \le i \ne j \le N} \mathrm{ES}_{\phi_i, \phi_j}(\mathbf{s}, \mathbf{a}) \right)$$

6:     Update policy with gradient ascent using

$$\nabla_\theta \frac{1}{|B|} \sum_{\mathbf{s} \in B} \left( \min_{j=1,\ldots,N} Q_{\phi_j}\left(\mathbf{s}, \tilde{\mathbf{a}}_\theta(\mathbf{s})\right) - \beta \log \pi_\theta\left(\tilde{\mathbf{a}}_\theta(\mathbf{s}) \mid \mathbf{s}\right) \right),$$

    where $\tilde{\mathbf{a}}_\theta(\mathbf{s})$ is a sample from $\pi_\theta(\cdot \mid \mathbf{s})$ which is differentiable w.r.t. $\theta$ via the reparametrization trick.
7:     Update target networks with $\phi_i' \leftarrow \rho\phi_i' + (1-\rho)\phi_i$

---

(Section 5.2) domains. We consider the following baselines: SAC, the backbone algorithm of our method, CQL, the previous state-of-the-art on the D4RL benchmark, REM [2], an offline RL method which utilized Q-network ensemble on discrete control environments, and BC, the behavior cloning method. We evaluate each method under the normalized average return metric where the average return is scaled such that 0 and 100 each equals the performance of a random policy and an online expert policy. In addition to the performance evaluation, we compare the computational cost of each method (Section 5.3). For the implementation details of our algorithm and the baselines, please refer to Appendix B and C. Also, we provide more experiments such as comparison with more baselines, CQL with $N$ Q-networks, and hyperparameter sensitivity from Appendix E to H. The code is available online[3].

## 5.1 Evaluation on D4RL MuJoCo Gym tasks

We first evaluate each method on D4RL MuJoCo Gym tasks which consist of three environments, halfcheetah, hopper, and walker2d, each with six datasets from different data-collecting policies. In detail, the considered policies are *random*: a uniform random policy, *expert*: a fully trained online expert, *medium*: a suboptimal policy with approximately 1/3 the performance of the expert, *medium-expert*: a mixture of medium and expert policies, *medium-replay*: the replay buffer of a policy trained up to the performance of the medium agent, and *full-replay*: the final replay buffer of the expert policy. Each dataset consists of 1M transitions except for medium-expert and medium-replay.

The experiment results in Table 1 show EDAC and SAC-$N$ both outperform or are competitive with the previous state-of-the-art on all of the tasks considered. Notably, the performance gap is especially high for random, medium, and medium-replay datasets, where the performances of the previous works are relatively low. Both the proposed methods achieve average normalized scores over 80, reducing the gap with the online expert by 40% compared to CQL. While the performance of EDAC is marginally better than the performance of SAC-$N$, EDAC achieves this result with a much smaller Q-ensemble size. As noted in Figure 5, on hopper tasks, SAC-$N$ requires 200 to 500 Q-networks, while EDAC requires less than 50.

Figure 6 compares the distance between the actions chosen by each method and the dataset actions. Concretely, we measure $\mathbb{E}_{(\mathbf{s}, \mathbf{a}) \sim \mathcal{D}, \hat{\mathbf{a}} \sim \pi_\theta(\cdot \mid \mathbf{s})} [\|\hat{\mathbf{a}} - \mathbf{a}\|_2^2]$ for EDAC, SAC-$N$, CQL, SAC-2, and a random

---

[3] `https://github.com/snu-mllab/EDAC`

policy on ∗-medium datasets. We find that our proposed methods choose from a more diverse range of actions compared to CQL. This shows the advantage of the uncertainty-based penalization which considers the prediction confidence other than penalizing all OOD actions.

Table 1: Normalized average returns on D4RL Gym tasks, averaged over 4 random seeds. CQL (Paper) denotes the results reported in the original paper.

| Task Name | BC | SAC | REM | CQL (Paper) | CQL (Reproduced) | SAC-$N$ (Ours) | EDAC (Ours) |
|---|---|---|---|---|---|---|---|
| halfcheetah-random | 2.2±0.0 | 29.7±1.4 | -0.8±1.1 | **35.4** | 31.3±3.5 | 28.0±0.9 | 28.4±1.0 |
| halfcheetah-medium | 43.2±0.6 | 55.2±27.8 | -0.8±1.3 | 44.4 | 46.9±0.4 | **67.5±1.2** | **65.9±0.6** |
| halfcheetah-expert | 91.8±1.5 | -0.8±1.8 | 4.1±5.7 | 104.8 | 97.3±1.1 | **105.2±2.6** | **106.8±3.4** |
| halfcheetah-medium-expert | 44.0±1.6 | 28.4±19.4 | 0.7±3.7 | 62.4 | 95.0±1.4 | **107.1±2.0** | **106.3±1.9** |
| halfcheetah-medium-replay | 37.6±2.1 | 0.8±1.0 | 6.6±11.0 | 46.2 | 45.3±0.3 | **63.9±0.8** | **61.3±1.9** |
| halfcheetah-full-replay | 62.9±0.8 | **86.8±1.0** | 27.8±35.4 | - | 76.9±0.9 | 84.5±1.2 | 84.6±0.9 |
| hopper-random | 3.7±0.6 | 9.9±1.5 | 3.4±2.2 | 10.8 | 5.3±0.6 | **31.3±0.0** | 25.3±10.4 |
| hopper-medium | 54.1±3.8 | 0.8±0.0 | 0.7±0.0 | 86.6 | 61.9±6.4 | **100.3±0.3** | **101.6±0.6** |
| hopper-expert | 107.7±9.7 | 0.7±0.0 | 0.8±0.0 | 109.9 | 106.5±9.1 | **110.3±0.3** | **110.1±0.1** |
| hopper-medium-expert | 53.9±4.7 | 0.7±0.0 | 0.8±0.0 | **111.0** | 96.9±15.1 | 110.1±0.3 | 110.7±0.1 |
| hopper-medium-replay | 16.6±4.8 | 7.4±0.5 | 27.5±15.2 | 48.6 | 86.3±7.3 | **101.8±0.5** | **101.0±0.5** |
| hopper-full-replay | 19.9±12.9 | 41.1±17.9 | 19.7±24.6 | - | 101.9±0.6 | **102.9±0.3** | **105.4±0.7** |
| walker2d-random | 1.3±0.1 | 0.9±0.8 | 6.9±8.3 | 7.0 | 5.4±1.7 | **21.7±0.0** | 16.6±7.0 |
| walker2d-medium | 70.9±11.0 | -0.3±0.4 | 0.2±0.7 | 74.5 | 79.5±3.2 | **87.9±0.2** | **92.5±0.8** |
| walker2d-expert | 108.7±0.2 | 0.7±0.3 | 1.0±2.3 | **121.6** | 109.3±0.1 | 107.4±2.4 | 115.1±1.9 |
| walker2d-medium-expert | 90.1±13.2 | 1.9±3.9 | -0.1±0.0 | 98.7 | 109.1±0.2 | **116.7±0.4** | **114.7±0.9** |
| walker2d-medium-replay | 20.3±9.8 | -0.4±0.3 | 12.5±6.2 | 32.6 | 76.8±10.0 | **78.7±0.7** | **87.1±2.3** |
| walker2d-full-replay | 68.8±17.7 | 27.9±47.3 | -0.2±0.3 | - | 94.2±1.9 | 94.6±0.5 | **99.8±0.7** |
| Average | 49.9 | 16.2 | 6.2 | - | 73.7 | **84.5** | **85.2** |

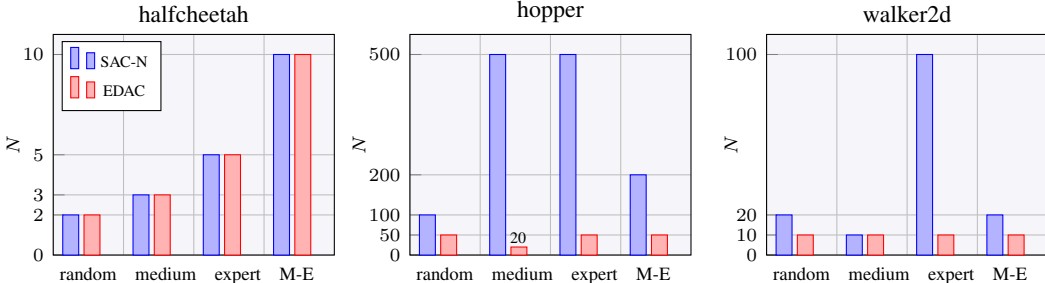

Figure 5: Minimum number of Q-ensembles ($N$) required to achieve the performance reported in Table 1. M-E denotes medium-expert. We omit the results of medium-replay and full-replay as SAC-$N$ already works well with a small number of ensembles (less than or equal to 5). For more details of the experiment, please refer to Appendix C.

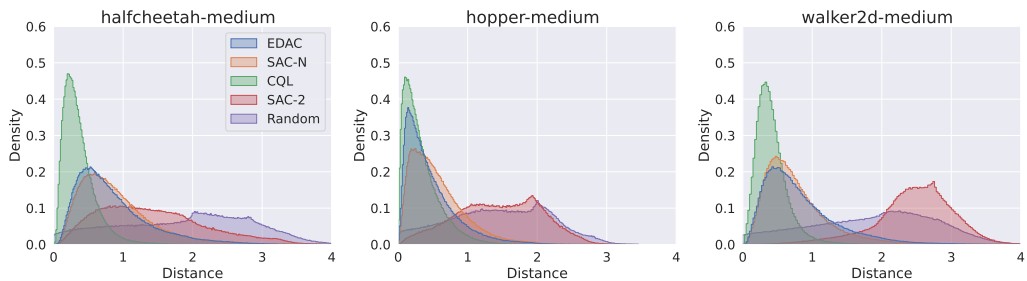

Figure 6: Histograms of the distances between the actions from each methods (EDAC, SAC-$N$, CQL, SAC-2, and a random policy) and the actions from the dataset. For more details of the experiment, please refer to Appendix C.

## 5.2 Evaluation on D4RL Adroit tasks

We also experiment on the more complex D4RL Adroit tasks that require controlling a 24-DoF robotic hand to perform tasks such as aligning a pen, hammering a nail, opening a door, or relocating a ball. We use two types of datasets for each environment: *human*, containing 25 trajectories of human demonstrations, and *cloned*, a 50-50 mixture between the demonstration data and the behavioral cloned policy on the demonstrations. Note that for the Adroit tasks, we could not reproduce the CQL results from the paper completely. For the detailed procedure of reproducing the results of CQL, please refer to Appendix D.

Table 2: Normalized average returns on D4RL Adroit tasks, averaged over 4 random seeds.

| Task Name | BC | SAC | REM | CQL (Paper) | CQL (Reproduced) | SAC-$N$ (Ours) | EDAC (Ours) |
|---|---|---|---|---|---|---|---|
| pen-human | 25.8±8.8 | 4.3±3.8 | 5.4±4.3 | **55.8** | 35.2±6.6 | 9.5±1.1 | 52.1±8.6 |
| hammer-human | 3.1±3.2 | 0.2±0.0 | 0.3±0.0 | 2.1 | 0.6±0.5 | 0.3±0.0 | 0.8±0.4 |
| door-human | 2.8±0.7 | -0.3±0.0 | -0.3±0.0 | 9.1 | 1.2±1.8 | -0.3±0.0 | **10.7±6.8** |
| relocate-human | 0.0±0.0 | -0.3±0.0 | -0.3±0.0 | 0.35 | 0.0±0.0 | -0.1±0.1 | 0.1±0.1 |
| pen-cloned | 38.3±11.9 | -0.8±3.2 | -1.0±0.1 | 40.3 | 27.2±11.3 | **64.1±8.7** | **68.2±7.3** |
| hammer-cloned | 0.7±0.3 | 0.1±0.1 | -0.3±0.0 | 5.7 | 1.4±2.1 | 0.2±0.2 | 0.3±0.0 |
| door-cloned | 0.0±0.0 | -0.3±0.1 | -0.3±0.0 | 3.5 | 2.4±2.4 | -0.3±0.0 | **9.6±8.3** |
| relocate-cloned | 0.1±0.0 | -0.1±0.1 | -0.2±0.2 | -0.1 | 0.0±0.0 | 0.0±0.0 | 0.0±0.0 |

The evaluation results are summarized in Table 2. For pen-∗ tasks, where the considered algorithms achieve meaningful performance, EDAC outperforms or matches with the previous state-of-the-art. Especially, for pen-cloned, both EDAC and SAC-$N$ achieve 75% higher score compared to CQL. Unlike the results from the Gym tasks, we find that SAC-$N$ falls behind in some datasets, for example, pen-human, which could in part due to the size of the dataset being exceptionally small (5000 transitions). However, our method with ensemble diversification successfully overcomes this difficulty.

## 5.3 Computational cost comparison

We compared the computational cost of our methods with vanilla SAC and CQL on hopper-medium-v2, where our methods require the largest number of Q-networks. For each method, we measure the runtime per training epoch (1000 gradient steps) along with GPU memory consumption. We run our experiments on a single machine with one RTX 3090 GPU and provide the results in Table 3.

As the result shows, our method EDAC runs faster than CQL with comparable memory consumption. Note that CQL is about twice as slower than vanilla SAC due to the

Table 3: Computational costs of each method.

| | Runtime (s/epoch) | GPU Mem. (GB) |
|---|---|---|
| **SAC** | 21.4 | 1.3 |
| **CQL** | 38.2 | 1.4 |
| **SAC**-500 | 44.1 | 5.1 |
| **EDAC** | 30.8 | 1.8 |

additional computations for Q-value regularization (e.g., dual update and approximate logsumexp via sampling). Meanwhile, the inference to the Q-network ensemble in SAC-$N$ and EDAC is embarrassingly parallelizable, minimizing the runtime increase with the number of Q-networks. Also, we emphasize that our gradient diversification term in Equation (4) has linear computational complexity, as we can reformulate the term using the sum of the gradients.

## 6 Related Works

**Model-free offline RL** A popular approach for offline RL is to regularize the learned policy to be close to the behavior policy where the offline dataset was collected. BCQ [11] uses a generative model to produce actions with high similarity to the dataset and trains a restricted policy to choose the best action from the neighborhood of the generated actions. Another line of work, such as BEAR [15] or BRAC [28], stabilizes policy learning by penalizing the divergence from the dataset measured by KL divergence or MMD. While these policy-constraint methods demonstrate high performance on

datasets from expert behavior policies, they fail to find optimal policies from datasets with suboptimal policies due to the strict policy constraints [9]. Also, these methods require an accurate estimation of the behavior policy, which might be difficult in complex settings with multiple behavior sources or high-dimensional environments. To address these issues, CQL [16] directly regularizes Q-functions by introducing a term that minimizes the Q-values for out-of-distribution actions and maximizes the Q-values for in-distribution actions. Without such explicit regularizations, REM [2] proposes to use a random convex combination of Q-network ensembles on environments with discrete action spaces [4].

**Estimation bias in Q-learning**   While Q-learning is one of the most popular algorithms in reinforcement learning, it suffers from overestimation bias due to the maximum operation $\max_{\mathbf{a}' \in \mathcal{A}} Q(\mathbf{s}', \mathbf{a}')$ used during Q-function updates [10, 25]. This overestimation bias, together with the bootstrapping, can lead to a catastrophic build-up of errors during the Q-learning process. To resolve this issue, TD3 [10] introduces a clipped version of Double Q-learning [25] that takes the minimum value of two critics. Subsequently, Maxmin Q-learning [17] theoretically shows that the overestimation bias can be controlled by the number of ensembles in the clipped Q-learning. The overestimation problem in Q-learning can be exacerbated in the offline setting since the extrapolation error cannot be corrected with further interactions with the environment, and existing offline RL algorithms handle the bias by introducing constrained policy optimization [11, 15] or conservative Q-learning frameworks [16].

**Uncertainty measures in RL**   Uncertainty estimates have been widely used in RL for various purposes including exploration, Q-learning, and planning. Bootstrapped DQN [21] leverages an ensemble of Q-functions to quantify the uncertainty of the Q-value, and utilizes it for efficient exploration. Following this work, the UCB exploration algorithm [5] constructs an upper confidence bound [3] of the Q-values using the empirical mean and standard deviation of Q-ensembles, which is used to promote efficient exploration by applying the principle of optimism in the face of uncertainty [7]. Osband et al. [22] proposes a randomly initialized Q-ensemble that reflects the concept of prior functions in Bayesian inference and Abbas et al. [1] introduces an uncertainty incorporated planning with imperfect models. The notion of uncertainty has also been considered in offline RL, mostly in the framework of model-based offline RL. Especially, MOPO [29] and MOReL [13] measure the uncertainty of the model's prediction to formulate an uncertainty-penalized policy optimization problem in the offline RL setting. These methods introduce an ensemble of dynamics models for the quantification of the uncertainty, whereas our work adopts an ensemble of Q-functions for uncertainty-aware Q-learning.

# 7   Conclusion

We have shown that clipped Q-learning can be efficiently leveraged to construct an uncertainty-based offline RL method that outperforms previous methods on various datasets. Based on this observation, we proposed Ensemble-Diversifying Actor-Critic (EDAC) that effectively reduces the required number of ensemble networks for quantifying and penalizing the epistemic uncertainty. Our method does not require any explicit estimation of the data collecting policy or sampling from the out-of-distribution data and respects the epistemic uncertainty of each data point during penalization. EDAC, while requiring up to 90% less number of ensemble networks compared to the vanilla Q-ensemble, exhibits state-of-the-art performance on various datasets.

# Acknowledgements

This work was supported in part by Samsung Advanced Institute of Technology, Samsung Electronics Co., Ltd., Institute of Information & Communications Technology Planning & Evaluation (IITP) grant funded by the Korea government (MSIT) (No. 2020-0-00882, (SW STAR LAB) Development of deployable learning intelligence via self-sustainable and trustworthy machine learning and No. 2019-0-01371, Development of brain-inspired AI with human-like intelligence), and Research Resettlement Fund for the new faculty of Seoul National University. This material is based upon work supported by the Air Force Office of Scientific Research under award number FA2386-20-1-4043. Hyun Oh Song is the corresponding author.

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
