# Supplementary Material for Uncertainty-Based Offline Reinforcement Learning with Diversified Q-Ensemble

## A  Ensemble gradient diversification

### A.1  Proofs

**Lemma 1.** *The total variance of the matrix* $\mathrm{Var}\left(\nabla_{\mathbf{a}} Q_{\phi_j}(\mathbf{s}, \mathbf{a})\right)$ *is equal to* $1 - \|\bar{q}\|_2^2$, *where* $\bar{q} = \frac{1}{N}\sum_{j=1}^{N} \nabla_{\mathbf{a}} Q_{\phi_j}(\mathbf{s}, \mathbf{a})$.

*Proof.* For simplicity, we denote $\nabla_{\mathbf{a}} Q_{\phi_j}(\mathbf{s}, \mathbf{a})$ by $q_j$ and their average by $\bar{q} = \frac{1}{N}\sum_j q_j$. Then, the total variance of the matrix, which is equivalent to the trace of the matrix by definition, formulates as below:

$$
\begin{aligned}
\mathrm{tr}\left(\mathrm{Var}\left(\nabla_{\mathbf{a}} Q_{\phi_j}(\mathbf{s}, \mathbf{a})\right)\right) &= \mathrm{tr}\left(\frac{1}{N}\sum_{j=1}^{N}(q_j - \bar{q})(q_j - \bar{q})^{\intercal}\right) \\
&= \frac{1}{N}\sum_{j=1}^{N}\mathrm{tr}\left((q_j - \bar{q})(q_j - \bar{q})^{\intercal}\right) \\
&= \frac{1}{N}\sum_{j=1}^{N}\mathrm{tr}\left((q_j - \bar{q})^{\intercal}(q_j - \bar{q})\right) & (\mathrm{tr}(AB) = \mathrm{tr}(BA)) \\
&= \frac{1}{N}\sum_{j=1}^{N}(q_j - \bar{q})^{\intercal}(q_j - \bar{q}) \\
&= \frac{1}{N}\sum_{j=1}^{N}\left(q_j^{\intercal}q_j - 2q_j^{\intercal}\bar{q} + \bar{q}^{\intercal}\bar{q}\right) \\
&= 1 - 2\left(\frac{1}{N}\sum_{j=1}^{N}q_j\right)^{\intercal}\bar{q} + \bar{q}^{\intercal}\bar{q} & (\|q_j\|_2 = 1) \\
&= 1 - \bar{q}^{\intercal}\bar{q} \\
&= 1 - \|\bar{q}\|_2^2.
\end{aligned}
$$

$\square$

**Proposition 1.** *Suppose* $Q_{\phi_j}(\mathbf{s}, \mathbf{a}) = Q(\mathbf{s}, \mathbf{a})$ *and* $Q_{\phi_j}(\mathbf{s}, \cdot)$ *is locally linear in the neighborhood of* $\mathbf{a}$ *for all* $j \in [N]$. *Let* $\lambda_{\min}$ *and* $\mathbf{w}_{\min}$ *be the smallest eigenvalue and the corresponding normalized eigenvector of the matrix* $\mathrm{Var}\left(\nabla_{\mathbf{a}} Q_{\phi_j}(\mathbf{s}, \mathbf{a})\right)$ *and* $\epsilon > 0$ *be the value such that* $\min_{i \neq j}\left\langle \nabla_{\mathbf{a}} Q_{\phi_i}(\mathbf{s}, \mathbf{a}), \nabla_{\mathbf{a}} Q_{\phi_j}(\mathbf{s}, \mathbf{a})\right\rangle = 1 - \epsilon$. *Then, the variance of the Q-values for an OOD*

*action in the neighborhood along the direction of $\mathbf{w}_{\min}$ is upper-bounded as follows:*

$$\mathrm{Var}\left(Q_{\phi_j}(\mathbf{s}, \mathbf{a} + k\mathbf{w}_{\min})\right) \leq \frac{1}{|\mathcal{A}|}\frac{N-1}{N}k^2\epsilon,$$

*where $|\mathcal{A}|$ is the action space dimension.*

*Proof.* We first prove that the smallest eigenvalue $\lambda_{\min}$ of $\mathrm{Var}\left(\nabla_{\mathbf{a}}Q_{\phi_j}(\mathbf{s}, \mathbf{a})\right)$ is upper-bounded by some constant multiple of $\epsilon$. For simplicity, we denote $\nabla_{\mathbf{a}}Q_{\phi_j}(\mathbf{s}, \mathbf{a})$ by $q_j$ and their average by $\bar{q} = \frac{1}{N}\sum_j q_j$. We first compute the norm of the average of the gradients, which can be expressed by

$$\begin{aligned}
\|\bar{q}\|_2^2 &= \langle \bar{q}, \bar{q} \rangle \\
&= \left\langle \frac{1}{N}\sum_{i=1}^{N} q_i, \frac{1}{N}\sum_{j=1}^{N} q_j \right\rangle \\
&= \frac{1}{N^2}\sum_{1 \leq i,j \leq N} \langle q_i, q_j \rangle \\
&= \frac{1}{N^2}\left(\sum_{j=1}^{N}\langle q_j, q_j \rangle + \sum_{1 \leq i \neq j \leq N} \langle q_i, q_j \rangle\right) \\
&\geq \frac{1}{N^2}\left(N + N(N-1)(1-\epsilon)\right) \\
&= 1 - \frac{(N-1)}{N}\epsilon.
\end{aligned}$$

By Lemma 1, the total variance of the matrix is less or equal to $\frac{N-1}{N}\epsilon$. Using the fact that the total variance is equivalent to the sum of the eigenvalues and the eigenvalues of a variance matrix is non-negative, we have

$$\begin{aligned}
\lambda_{\min} &\leq \frac{1}{|\mathcal{A}|}\sum_{j=1}^{|\mathcal{A}|}\lambda_j \\
&= \frac{1}{|\mathcal{A}|}\mathrm{tr}\left(\mathrm{Var}\left(\nabla_{\mathbf{a}}Q_{\phi_j}(\mathbf{s}, \mathbf{a})\right)\right) \\
&\leq \frac{1}{|\mathcal{A}|}\frac{N-1}{N}\epsilon,
\end{aligned} \tag{1}$$

where $\lambda_1, \ldots, \lambda_{|\mathcal{A}|}$ are the eigenvalues of $\mathrm{Var}\left(\nabla_{\mathbf{a}}Q_{\phi_j}(\mathbf{s}, \mathbf{a})\right)$.

Note that, using the fact that the Q-values coincide at the action $\mathbf{a}$ and the local linearity of the Q-functions, we have derived

$$\mathrm{Var}(Q_{\phi_j}(\mathbf{s}, \mathbf{a} + k\mathbf{w})) = k^2\mathbf{w}^{\mathsf{T}}\mathrm{Var}\left(\nabla_{\mathbf{a}}Q_{\phi_j}(\mathbf{s}, \mathbf{a})\right)\mathbf{w}. \tag{2}$$

Plugging $\mathbf{w} = \mathbf{w}_{\min}$ in Equation (2) and using Equation (1), we have

$$\begin{aligned}
\mathrm{Var}(Q_{\phi_j}(\mathbf{s}, \mathbf{a} + k\mathbf{w}_{\min})) &= k^2\mathbf{w}_{\min}^{\mathsf{T}}\mathrm{Var}\left(\nabla_{\mathbf{a}}Q_{\phi_j}(\mathbf{s}, \mathbf{a})\right)\mathbf{w}_{\min} \\
&= k^2\lambda_{\min} \\
&\leq \frac{1}{|\mathcal{A}|}\frac{N-1}{N}k^2\epsilon.
\end{aligned}$$

$\square$

.

## A.2 Relationship between maximizing the total variance and maximizing the smallest eigenvalue

As we have shown in Section 4, maximizing the total variance of the matrix $\mathrm{Var}\left(\nabla_{\mathbf{a}}Q_{\phi_i}(\mathbf{s}, \mathbf{a})\right)$ is equivalent to minimizing the cosine similarity of all distinct pairs of the gradients $\nabla_{\mathbf{a}}Q_{\phi_i}(\mathbf{s}, \mathbf{a})$,

which makes the gradients uniformly distributed on the unit sphere $S^{|\mathcal{A}|-1}$. Therefore, if the trace is sufficiently maximized, then we can see $\text{Var}\left(\nabla_{\mathbf{a}}Q_{\phi_i}(\mathbf{s},\mathbf{a})\right)$ as a sample variance matrix of a uniform spherical distribution. It can be easily proved that the variance matrix of a uniform distribution on is $\frac{1}{|\mathcal{A}|}I$, whose all eigenvalues are equal to $\frac{1}{|\mathcal{A}|}$, by Proposition 2.

**Proposition 2.** *The variance matrix of the uniform spherical distribution $X \sim \mathcal{U}(S^{n-1})$ is $\frac{1}{n}I$.*

*Proof.* Let $X = (X_1, \ldots, X_n)$. Then $X_{-i} = (X_1, \ldots, -X_i, \ldots, X_n)$ is also from the uniform spherical distribution. Therefore, we have $\mathbb{E}[X_i] = \mathbb{E}[-X_i] = 0$ and $\mathbb{E}[X_i X_j] = \mathbb{E}[-X_i X_j] = 0$, $\forall i \neq j$. For the diagonal entries of the variance matrix, we have $\mathbb{E}[\sum_{i=1}^n X_i^2] = \sum_{i=1}^n \mathbb{E}[X_i^2] = 1$ by the definition of the spherical distribution and $\mathbb{E}[X_i^2] = \mathbb{E}[X_j^2]$ by the symmetry of the distribution. Therefore, we have $\mathbb{E}[X_i^2] = \frac{1}{n}$ and $\text{Var}(X) = \frac{1}{n}I$. $\qquad\square$

Note that the smallest eigenvalue of $\text{Var}\left(\nabla_{\mathbf{a}}Q_{\phi_i}(\mathbf{s},\mathbf{a})\right)$ is less or equal to $\frac{1}{|\mathcal{A}|}$, since the total variance is upper-bounded by 1 due to Lemma 1. Therefore, as the number of Q-ensembles goes to infinity, $\text{Var}\left(\nabla_{\mathbf{a}}Q_{\phi_i}(\mathbf{s},\mathbf{a})\right)$ converges to $\frac{1}{|\mathcal{A}|}I$, attaining the maximum value for the smallest eigenvalue.

# B    Implementation details

**SAC**    We use the SAC implementation built on rlkit[1]. We use its default parameters except for increasing the number of layers for both the policy network and the Q-function networks from 2 to 3, following the protocol of CQL.

**REM**    We implement a continuous control version of REM on top of SAC by modifying the Bellman residual term to

$$\min_{\phi} \ \mathbb{E}_{\mathbf{s},\mathbf{a},\mathbf{s}' \sim \mathcal{D}, \xi \sim P_{\Delta}} \left[ \left( \left( \sum_{j=1}^N \xi_j Q_{\phi_j}(\mathbf{s},\mathbf{a}) \right) - \left( r(\mathbf{s},\mathbf{a}) + \gamma \, \mathbb{E}_{\mathbf{a}' \sim \pi_\theta(\cdot|\mathbf{s}')} \left[ \sum_{j=1}^N \xi_j Q_{\phi_j'}\left(\mathbf{s}',\mathbf{a}'\right) \right] \right) \right)^2 \right],$$

where $P_{\Delta}$ represents a probability distribution over the standard $(N-1)$-simplex $\Delta^{N-1} = \{\xi \in \mathbb{R}^N : \xi_1 + \xi_2 + \cdots + \xi_N = 1, \ \xi_n \geq 0, \ n = 1, \ldots, N\}$. Following the original REM paper, we use a simple probability distribution: $\xi_n = \xi_n' / \sum_k \xi_k'$, where $\xi_k' \sim U(0,1)$ for $k = 1, \ldots, N$. For a fair comparison with our ensemble algorithms, we sweep the ensemble size $N$ within $\{2, 5, 10, 20, 50, 100, 200, 500, 1000\}$ and report the best number.

**CQL**    We use the official implementation by the authors[2]. For MuJoCo Gym tasks, the recommended hyperparameters in the codebase differ from the original paper due to the updates in the D4RL datasets. We tried both versions of hyperparameter settings and found the codebase version outperforms the paper version while matching the numbers in the paper reasonably well. Therefore, we follow the guidelines from the official code and use the fixed $\alpha$ version, searching for the parameters within $\alpha \in \{5, 10\}$ and policy learning rate $\in \{1e-4, 3e-4\}$. We chose $\alpha = 10.0$ with policy learning rate $= 1e-4$ as the default as it gives the best results in most of the datasets. However, we use the dual gradient descent version with $\tau = 10.0$ and policy learning rate$= 1e-4$ on some datasets, such as halfcheetah-random, since the fixed $\alpha$ version could not reproduce the results from the paper on those datasets. For the Adroit tasks, the codebase does not provide separate guidelines, and we use the hyperparameters listed in the paper.

**SAC-$N$ (Ours)**    We keep the default setting from the SAC experiments other than the ensemble size $N$. On halfcheetah and walker2d environments, we tune $N$ in the range of $\{5, 10, 20\}$ except for walker2d-expert, which requires up to $N = 100$. For hopper, we tune within $N \in \{100, 200, 500, 1000\}$. The hyperparameters selected are listed in Table 1. As we noted in our main paper's Figure 5, some datasets can be dealt with less $N$ (*e.g.*, ∗-replay). However, we tried to keep the hyperparameters within an environment consistent in order to reduce hyperparameter sensitivity. Also, we find reward normalization to help stabilize the uncertainty penalization in some of the

---

[1]`https://github.com/vitchyr/rlkit`
[2]`https://github.com/aviralkumar2907/CQL`

datasets (*e.g.*, walker2d-expert). For Adroit tasks, we sweep $N$ in the range of $\{20, 50, 100, 200\}$ and adopt max Q backup from CQL for training stability. We report the selected $N$ in Table 2.

**EDAC (Ours)** For Mujoco Gym tasks, we tune the ensemble size $N$ within the range of $\{10, 20, 50\}$ and the weight of the ensemble gradient diversity term $\eta$ within $\{0.0, 1.0, 5.0\}$. Note that we use the same $N$ on each environment. For Adroit tasks, we sweep the parameters on $N \in \{20, 50, 100\}$ and $\eta \in \{100, 200, 500, 1000\}$ except for pen-cloned, which uses $\eta = 10.0$. While we can also achieve competitive performance on pen-cloned with larger $\eta$, we found lower $\eta$ helps to mitigate the performance degradation on further training steps. As from SAC-$N$, we use max Q backup on some of the datasets (pen-cloned). The selected $N$ and $\eta$ for each environment are listed in Table 1 and Table 2, respectively.

Table 1: Hyperparameters used in the D4RL MuJoCo Gym experiments.

| Task Name | SAC-$N$ ($N$) | EDAC ($N$, $\eta$) |
|---|---|---|
| halfcheetah-random | 10 | 10, 0.0 |
| halfcheetah-medium | 10 | 10, 1.0 |
| halfcheetah-expert | 10 | 10, 1.0 |
| halfcheetah-medium-expert | 10 | 10, 5.0 |
| halfcheetah-medium-replay | 10 | 10, 1.0 |
| halfcheetah-full-replay | 10 | 10, 1.0 |
| hopper-random | 500 | 50, 0.0 |
| hopper-medium | 500 | 50, 1.0 |
| hopper-expert | 500 | 50, 1.0 |
| hopper-medium-expert | 200 | 50, 1.0 |
| hopper-medium-replay | 200 | 50, 1.0 |
| hopper-full-replay | 200 | 50, 1.0 |
| walker2d-random | 20 | 10, 1.0 |
| walker2d-medium | 20 | 10, 1.0 |
| walker2d-expert | 100 | 10, 5.0 |
| walker2d-medium-expert | 20 | 10, 5.0 |
| walker2d-medium-replay | 20 | 10, 1.0 |
| walker2d-full-replay | 20 | 10, 1.0 |

Table 2: Hyperparameters used in the D4RL Adroit experiments.

| Task Name | SAC-$N$ ($N$) | EDAC ($N$, $\eta$) |
|---|---|---|
| pen-human | 100 | 20, 1000.0 |
| pen-cloned | 100 | 20, 10.0 |
| hammer-human | 100 | 50, 200.0 |
| hammer-cloned | 100 | 50, 200.0 |
| door-human | 100 | 50, 200.0 |
| door-cloned | 100 | 50, 200.0 |
| relocate-human | 100 | 50, 200.0 |
| relocate-cloned | 100 | 50, 200.0 |

## C  Experimental settings

**MuJoCo Gym** We use the v2 version of each dataset (*e.g.*, halfcheetah-random-v2) which fixes some of the bugs from the previous versions. We run each algorithm for 3 million training steps and report the normalized average return of each policy. While the CQL paper originally used 1 million steps, we found increasing this to 3 million helps the algorithms to converge on more complex datasets such as ∗-medium-expert.

**Adroit**   We use the v1 version of each dataset and normalize the rewards. As we will discuss in Appendix D, the performance of the baseline algorithm CQL degrades after some steps of training. Therefore, for a fair comparison, we run each algorithm for 200,000 steps and report the normalized average return.

**Measuring minimum required Q-ensembles (main paper Figure 5)**   To check the minimum required number of Q-ensembles for each dataset, we sweep $N$ within the range of $\{2, 3, 5, 10, 20, 50, 100, 200, 500, 1000\}$ and report the minimum $N$ that achieves the performance similar to Table 1 of the main paper. We find EDAC successes to reduce the required $N$ significantly when the original requirement is high (*e.g.*, hopper, walker2d-expert).

**Action distance histograms (main paper Figure 6)**   To draw the histogram, we sample 500,000 random $(\mathbf{s}, \mathbf{a})$ pairs from each dataset and measure the $\ell_2$ distance between the action sampled from each policy after full training and the dataset action.

# D   Reproducing CQL in Adroit

Since the pen-∗ tasks are where the considered algorithms show meaningful performance, we focused on reproducing the reported results for those tasks. After running CQL with the parameters given in the original paper, we found that the performance of CQL degrades after about 200,000 steps, as shown in Figure 1. While we are not sure of the cause of this performance gap, it could be due to the difference in the `min_q_weight` parameter setting, which was not specified in the original paper, or a minor modification we applied to the code to fix the backpropagation issue[3]. Meanwhile, for a fair comparison, on Adroit we chose to use early-stopping and train each algorithm for 200,000 steps. Also, we include the reported CQL numbers for all experiments.

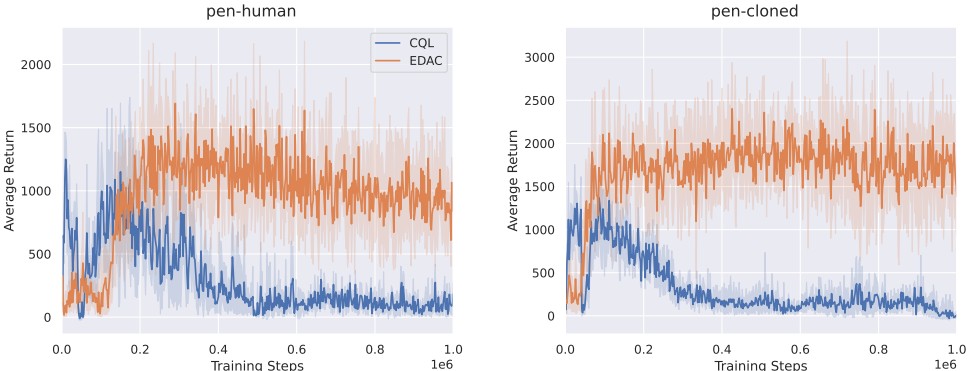

Figure 1: Performance of EDAC and CQL on pen-∗ datasets. 'Average Return' denotes the undiscounted return of each policy on evaluation. Results averaged over 4 seeds.

# E   Comparison with more baselines

We additionally compared our methods with more baselines on D4RL Gym datasets. First, we add comparisons with some of the well-known offline RL methods, BCQ [1], BEAR [3], BRAC [4], and MOReL [2]. Also, we include the results of UWAC [5], a concurrent work that also utilizes Q-value uncertainty. We reproduced all the methods by following the hyperparameter search procedure listed in each paper and selected the best results. We report the normalized average return results in Table 3.

---

[3]`https://github.com/aviralkumar2907/CQL/issues/5`

Table 3: Extended results of normalized average returns on D4RL Gym tasks, averaged over 4 random seeds. CQL (Paper) denotes the results reported in the original paper.

| Task Name | BC | SAC | REM | BCQ | BEAR | BRAC | MOReL | UWAC | CQL (Paper) | CQL (Reproduced) | SAC-$N$ (Ours) | EDAC (Ours) |
|---|---|---|---|---|---|---|---|---|---|---|---|---|
| halfcheetah-random | 2.2±0.0 | 29.7±1.4 | -0.8±1.1 | 2.2±0.0 | 12.6±1.0 | 24.3±0.7 | **38.9±1.8** | 2.3±0.0 | 35.4 | 31.3±3.5 | 28.0±0.9 | 28.4±1.0 |
| halfcheetah-medium | 43.2±0.6 | 55.2±27.8 | -0.8±1.3 | 46.6±0.4 | 42.8±0.1 | 51.9±0.3 | 60.7±4.4 | 43.7±0.4 | 44.4 | 46.9±0.4 | **67.5±1.2** | 65.9±0.6 |
| halfcheetah-expert | 91.8±1.5 | -0.8±1.8 | 4.1±5.7 | 89.9±9.6 | 92.6±0.6 | 39.0±13.8 | 8.4±11.8 | 94.7±1.1 | 104.8 | 97.3±1.1 | 105.2±2.6 | **106.8±3.4** |
| halfcheetah-medium-expert | 44.0±1.6 | 28.4±19.4 | 0.7±3.7 | 95.4±2.0 | 45.7±4.2 | 52.3±0.1 | 80.4±11.7 | 47.0±6.0 | 62.4 | 95.0±1.4 | **107.1±2.0** | 106.3±1.9 |
| halfcheetah-medium-replay | 37.6±2.1 | 0.8±1.0 | 6.6±11.0 | 42.2±0.9 | 39.4±0.8 | 48.6±0.4 | 44.5±5.6 | 38.9±1.1 | 46.2 | 45.3±0.3 | **63.9±0.8** | 61.3±1.9 |
| halfcheetah-full-replay | 62.9±0.8 | **86.8±1.0** | 27.8±35.4 | 69.5±4.0 | 60.1±3.2 | 78.0±0.7 | 70.1±5.1 | 65.1±0.5 | - | 76.9±0.9 | 84.5±1.2 | 84.6±0.9 |
| hopper-random | 3.7±0.6 | 9.9±1.5 | 3.4±2.2 | 7.8±0.6 | 3.6±3.6 | 8.1±0.6 | **38.1±10.1** | 2.6±0.3 | 10.8 | 5.3±0.6 | 31.3±0.0 | 25.3±10.4 |
| hopper-medium | 54.1±3.8 | 0.8±0.0 | 0.7±0.0 | 59.4±8.3 | 55.3±3.2 | 77.8±6.1 | 84.0±17.0 | 52.6±4.0 | 86.6 | 61.9±6.4 | 100.3±0.3 | **101.6±0.6** |
| hopper-expert | 107.7±9.7 | 0.7±0.0 | 0.8±0.0 | 109±4.0 | 39.4±20.5 | 78.1±52.3 | 80.4±34.9 | **111.0±0.8** | 109.9 | 106.5±9.1 | 110.3±0.3 | 110.1±0.1 |
| hopper-medium-expert | 53.9±4.7 | 0.7±0.0 | 0.8±0.0 | 106.9±5.0 | 66.2±8.5 | 81.3±8.0 | 105.6±8.2 | 54.8±3.2 | **111.0** | 96.9±15.1 | 110.1±0.3 | 110.7±0.1 |
| hopper-medium-replay | 16.6±4.8 | 7.4±0.5 | 27.5±15.2 | 60.9±14.7 | 57.7±16.5 | 62.7±30.4 | 81.8±17.0 | 31.1±14.8 | 48.6 | 86.3±7.3 | **101.8±0.5** | 101.0±0.5 |
| hopper-full-replay | 19.9±12.9 | 41.1±17.9 | 19.7±24.6 | 46.6±13.0 | 54.0±24.0 | **107.4±0.5** | 94.4±20.5 | 21.9±8.4 | - | 101.9±0.6 | 102.9±0.3 | 105.4±0.7 |
| walker2d-random | 1.3±0.1 | 0.9±0.8 | 6.9±8.3 | 4.9±0.1 | 4.3±1.2 | 1.3±1.4 | 16.0±7.7 | 1.5±0.3 | 7.0 | 5.4±1.7 | **21.7±0.0** | 16.6±7.0 |
| walker2d-medium | 70.9±11.0 | -0.3±0.2 | 0.2±0.7 | 71.8±7.2 | 59.8±40.0 | 59.7±39.9 | 72.8±11.9 | 66.0±9.0 | 74.5 | 79.5±3.2 | 87.9±0.2 | **92.5±0.8** |
| walker2d-expert | 108.7±0.2 | 0.7±0.3 | 1.0±2.3 | 106.3±5.0 | 110.1±0.6 | 55.2±62.2 | 62.6±29.9 | 108.4±0.5 | **121.6** | 109.3±0.1 | 107.4±2.4 | 115.1±1.9 |
| walker2d-medium-expert | 90.1±13.2 | 1.9±3.9 | -0.1±0.0 | 107.7±3.8 | 107.0±2.9 | 9.3±18.9 | 107.5±5.6 | 85.7±14.0 | 98.7 | 109.1±0.2 | **116.7±0.4** | 114.7±0.9 |
| walker2d-medium-replay | 20.3±9.8 | -0.4±0.3 | 12.5±6.2 | 57.0±9.6 | 12.2±4.7 | 40.1±47.9 | 40.8±20.4 | 27.1±9.6 | 32.6 | 76.8±10.0 | 78.7±0.7 | **87.1±2.3** |
| walker2d-full-replay | 68.8±17.7 | 27.9±47.3 | -0.2±0.3 | 71.0±21.8 | 79.6±15.6 | 96.9±2.2 | 84.8±13.1 | 60.7±15.6 | - | 94.2±1.9 | 94.6±0.5 | **99.8±0.7** |
| Average | 49.9 | 16.2 | 6.2 | 64.2 | 52.4 | 54.0 | 65.1 | 50.8 | - | 73.7 | 84.5 | 85.2 |

The results show our methods outperform all the baseline methods on most of the datasets considered. Also, we reiterate that while the performance of EDAC is marginally better than SAC-$N$, EDAC achieves this result with a much smaller Q-ensemble size.

# F CQL with $N$ Q-networks

Since other offline RL methods may also benefit from larger $N$ or ensemble diversification, here we evaluate CQL-$N$ and CQL with ensemble diversification for ablation. For CQL-$N$, we tried $N \in \{2, 5, 10, 50, 100\}$, where $N = 2$ denotes the original version of CQL. For CQL with ensemble diversification, we added our diversification term to the CQL loss function and swept the coefficient $\eta$ in the range of $\{0.5, 1.0, 5.0\}$, which is the same range used in EDAC. The normalized return evaluation results on D4RL Gym $*$-medium datasets are shown in Table 4.

Table 4: Performance of CQL with $N$ Q-networks on D4RL Gym $*$-medium datasets.

| | | halfcheetah-medium | hopper-medium | walker2-medium |
|---|---|---|---|---|
| **CQL-$N$** | $N = 2$ | 46.9±0.4 | 61.9±6.4 | 79.5±3.2 |
| | $N = 5$ | 47.1±0.3 | 61.6±6.0 | 80.8±4.9 |
| | $N = 10$ | 45.9±0.3 | 60.1±4.8 | 70.9±0.9 |
| | $N = 50$ | 44.2±0.4 | 54.3±2.0 | 69.4±0.0 |
| | $N = 100$ | 43.7±0.2 | 43.7±0.8 | 71.3±3.8 |
| **CQL w/ diversification** | $\eta = 0.5$ | 46.5±0.4 | 65.8±11.2 | 82.2±0.6 |
| | $\eta = 1.0$ | 47.2±0.1 | 69.2±8.8 | 80.5±3.1 |
| | $\eta = 5.0$ | 47.4±0.5 | 60.9±3.2 | 82.1±1.2 |
| **SAC-$N$** | | **67.5±1.2** | **100.3±0.3** | **87.9±0.2** |
| **EDAC** | | **65.9±1.6** | **101.6±0.6** | **92.5±0.8** |

We observe that even though increasing the number of Q-networks or applying gradient diversification do help CQL on some of the datasets, the improved performance still falls far behind our methods (SAC-$N$, EDAC).

# G Comparison to variance regularization

In this section, we compare EDAC with increasing the variance of the Q-estimates for in-distribution actions, which is another possible option for ensemble diversification. Table 5 shows the average return and the Q-value estimation statistics on the walker2d-expert dataset when using the Q-estimate variance regularizer, compared to EDAC. *Var reg* adds to SAC-$N$ a regularizing term that explicitly increases the variance of the Q-estimates, weighted by a coefficient $c$. *Q Avg* denotes the estimated Q-values of each model in evaluation. *Q Std* means the standard deviation of Q-estimates from a

ensemble on the given actions. *Q Std gap* means the gap of standard deviations from behavior and random actions.

Table 5: Comparison of EDAC with Q-estimate variance regularization on walker2d-expert dataset. Same number of Q-networks is used for all methods.

|  |  | Return | Q Avg | Q Std (behavior action) | Q Std (random action) | Q Std gap |
|---|---|---|---|---|---|---|
| **Var reg** | $c = 50$ | 511 | overflow | N/A | N/A | N/A |
|  | $c = 100$ | 20 | -95 | 5.3 | 7 | 1.7 |
|  | $c = 200$ | 368 | -929 | 10.6 | 15.1 | 4.5 |
| **EDAC** |  | 5236 | 392 | 1.2 | 10.5 | 9.3 |

On the walker2d-expert dataset, adding the variance-enhancing regularizer either leads to two results: (1) Exploding Q-values when the regularization is not strong ($c = 50$) or (2) severe Q-value underestimation when the regularization is stronger ($c = 100, 200$). The reason behind these two extreme modes is that the gap of the Q-estimate variance between behavior actions and OOD actions, which is crucial for conservative learning, increases much slower than the absolute increase of the Q-estimate variances. For example, on $c = 200$, the Q-estimate Std gap is 4.5. This gap is about half of EDAC, whereas the absolute Q-estimate Stds on both actions are much higher. In EDAC, the variance of Q-estimates on behavior actions remains small even though the OOD actions are sufficiently penalized, as we only diversify the Q-networks' gradients instead of the Q-values themselves.

## H  Hyperparameter sensitivity

To measure the hyperparameter sensitivity of EDAC, we sweep the weight of the gradient diversification term $\eta$ in the range of {0.0, 0.5, 1.0, 2.0, 5.0} on the hopper datasets, fixing the number of Q-networks to $N = 50$, and present the results in Table 6.

Table 6: Performance of EDAC over various $\eta$ on the D4RL Gym hopper datasets.

| Dataset type | $\eta = 0.0$ | $\eta = 0.5$ | $\eta = 1.0$ | $\eta = 2.0$ | $\eta = 5.0$ |
|---|---|---|---|---|---|
| random | 25.3±10.4 | 9.3±6.2 | 6.7±0.8 | 3.8±1.5 | 1.9±0.8 |
| medium | 7.3±0.1 | 102.2±0,4 | 101.6±0.5 | 94.5±12.4 | 75.5±24.1 |
| expert | 2.3±0.1 | 110.3±0.2 | 110.1±0.1 | 109.8±0.2 | 109.9±0.2 |
| medium-expert | 46.9±33.0 | 103.8±12.4 | 110.7±0.1 | 109.8±0.2 | 109.8±0.2 |
| medium-replay | 100.9±0.4 | 100.3±0.8 | 101.0±0.4 | 100.2±0.5 | 20.6±0.7 |
| full-replay | 105.6±0.4 | 104.9±0.5 | 4105.4±0.6 | 104.0±0.2 | 106.3±0.9 |

The results show that except for the random dataset, there exists a large well of hyperparameters where EDAC achieves expert-level performance. We also observe that increasing $\eta$ sometimes degrades the performance on random, medium, and medium-replay datasets which contain trajectories drawn from suboptimal policies. Intuitively, the gradient diversification term induces the learned policy to favor in-distribution actions over OOD actions. Therefore, increasing $\eta$ can lead to a more conservative policy, which is undesirable if the behavior policy is suboptimal.