# OpenReview forum: "Uncertainty-Based Offline Reinforcement Learning with Diversified Q-Ensemble"
_NeurIPS.cc/2021/Conference — NeurIPS 2021 Poster_

### Official Review · Reviewer_MGH5 · 2021-07-12

**Rating:** 6
**Confidence:** 4

**Summary:**

This paper proposes two novel approaches to address the overestimation of OOD action value in offline RL, increasing the number of critics in the online RL algorithm (SAC-N), and forcing each ensemble to estimate diverse values with gradient similarity minimization term (EDAC). Both of them outperform the baseline methods (BC, SAC, REM, and CQL) in D4RL benchmarks (MuJoCo and Adroit tasks).

**Limitations And Societal Impact:**

## Limitation
The proposed methods require a large number of Q-networks (up to 500 in SAC-N, even 50 in EDAC), which seems sometimes an infeasible approach since previous methods typically use only two.  I think such computational inefficiency is an obvious limitation and should be described (although there is a trade-off between the performance and costs).

## Societal Impact
N/A

**Main Review:**

## Summary
The authors propose a simple solution to handle the uncertainty in offline RL, increasing the number of Q ensembles and diversifying each estimate.
However, these enormous amounts of Q-functions and computation of gradient similarity seem to require a huge amount of computational costs, which might be an intractable solution. Also, some ablations might be missed. Considering these aspects, I lean towards rejection.


## Pros
- This paper is well written and easy to follow.
- The idea of diversifying Q ensemble estimates seems a simple yet effective approach.
- The experimental results seem good in D4RL MuJoCo and Adroit tasks over popular benchmark methods (e.g. CQL).


## Cons
- The proposed methods require a large number of Q-networks (up to 500 in SAC-N, 50 in EDAC), which seems sometimes an intractable approach. Also, I guess the gradient similarity term (Algorithm 1, L5) requires a lot of computational costs.
- In checklist 3 (d), the authors say the amount of computation is described in Appendix B, but I can't found it. Could you describe the computational burden (or point out the line number where you mentioned it)?
- Popular offline methods like BCQ, BEAR, BRAC, and CQL, also employ a clipped double Q-learning. The ablation of (Offline RL)-N, or offline RL w/ diversification might be missed, which might lead to better performance than SAC-N or EDAC.
- Figure 6 shows the average action distance to the datasets. I'm curious about the histogram of standard SAC (SAC-2). I guess that it might look the same as SAC-N or EDAC ones. If so, the performance gain of EDAC or EDAC might come from a different perspective.
- Eq. 5 shows an example of that variance of critics are bounded under some conditions and degrade the ability to penalize OOD actions, which motivate the diversification of the gradient of Q function. I think that Eq. 5 also holds in EDAC. Could the authors provide a tighter bound or intuitive interpretation?
- As for the related work, the authors can include EMaQ [Ghasemipour et al. 2021], which suggests that Q ensembles is an effective approach in offline RL, and UWAC [Wu et al. 2021], which proposes uncertainty-based regularization in offline RL. It may better to address the clarification of the connection to such works. If the code is available, the authors could take it into consideration to compare them empiricaly.
-  In appendix B, Table 1 shows that EDAC sometimes chooses $\eta$=0, which seems to reduce to SAC-N. I wonder this implies that the gradient correction in EDAC might be unstable and highly dependent on the parameter choices.
- (Minor) L77-79 argues that the methods that require an estimated behavior policy such as BCQ, BEAR, and BRAC just work in "simple environments". However, the benchmark environments used in those papers are the same as this paper, MuJoCo locomotion tasks, and they show decent success. I'm not sure this expression is a proper one.
- (Minor) L82-84 argues that CQL loss requires an appropriate action distribution and sampling, and it causes computational inefficiency. However, I wonder the proposed methods (SAC-N, and EDAC) also requires a lot of sampling procedure during training (Algorithm 1 L4) since the number of Q networks is large. I think it also causes computational inefficiency.
- (Minor) Notation of $\mu$ seems to be used in a different context (in L84, and Eq. 3).

### Reference
Ghasemipour, S. K. S., Schuurmans, D., & Gu, S. S. (2021). Emaq: Expected-max q-learning operator for simple yet effective offline and online rl. In International Conference on Machine Learning.

Wu, Y., Zhai, S., Srivastava, N., Susskind, J., Zhang, J., Salakhutdinov, R., & Goh, H. (2021). Uncertainty Weighted Actor-Critic for Offline Reinforcement Learning. In International Conference on Machine Learning.


**Time Spent Reviewing:**

10 hours

---

> ### Author Response · Authors · 2021-08-10
> **Response to Reviewer MGH5**
>
> Thank you for your encouraging comments and constructive feedback. We would like to address the reviewer’s comments below.
>
> **1. The computational cost of the proposed algorithm.**
>
> We apologize for the mistake. The computational cost discussion should have been included in our appendix. Please refer to Common response 2, where we compared the computational cost of our algorithm with CQL. In short, our method EDAC runs faster than CQL, with comparable memory consumption. This result is because the inference to the Q-networks is embarrassingly parallelizable, which minimizes the increase of runtime with the number of Q-networks. We will include these results and related discussions in our revised version.
>
> **2. The ablation of (Offline RL)-$N$, or offline RL w/ diversification might be missed.**
>
> We implemented CQL-$N$ and CQL w/ diversification on top of the CQL algorithm, and provide the normalized return evaluation results below. For CQL-$N$, we tried $N \\in \\{2, 5, 10, 50, 100\\}$, where $N=2$ denotes the original version of CQL. For CQL w/ diversification, we added our diversification term to the CQL loss function and swept the coefficient $\\eta$ in the range of $\\{0.5, 1.0, 5.0\\}$, which is the same with EDAC.
>
> |  |  | halfcheetah-medium | hopper-medium | walker2d-medium |
> |---|---|---:|---:|---:|
> | CQL-$N$ | $N=2$ | 46.9±0.4 | 61.9±6.4 | 79.5±3.2 |
> |  | $N=5$ | 47.1±0.3 | 61.6±6.0 | 80.8±4.9 |
> |  | $N=10$ | 45.9±0.3 | 60.1±4.8 | 70.9±0.9 |
> |  | $N=50$ | 44.2±0.4 | 54.3±2.0 | 69.4±0.0 |
> |  | $N=100$ | 43.7±0.2 | 43.7±0.8 | 71.3±3.8 |
> | CQL w/ diversification | $\eta=0.5$ | 46.5±0.4 | 65.8±11.2 | 82.2±0.6 |
> |  | $\\eta=1.0$ | 47.2±0.1 | 69.2±8.8 | 80.5±3.1 |
> |  | $\\eta=5.0$ | 47.4±0.5 | 60.9±3.2 | 82.1±1.2 |
> | SAC-$N$ |  | **67.5±1.2** | **100.3±0.3** | **87.9±0.2** |
> | EDAC |  | **65.9±1.6** | **101.6±0.6** | **92.5±0.8** |
>
> Even though increasing the number of Q-networks or applying gradient diversification do help CQL on some of the datasets, the improved performance still falls far behind our methods (SAC-$N$, EDAC). We will include these ablation results in the revised version.
>
> **3. Action distance histogram of SAC-2 might be similar to SAC-$N$ or EDAC.**
>
> We provide the action distance histogram including SAC-2 in the following link: https://drive.google.com/drive/folders/1dKhb_1t5MUcggyjLRomqe8fjFS0oDCs1?usp=sharing. We find that action samples from SAC-2 are much further away from the behavior actions than SAC-$N$ or EDAC. This difference can be due to the weaker OOD action penalization, which leads to erroneously preferring OOD actions.
>
> **4. Eq. 5 also holds in EDAC. Could the authors provide a tighter bound or intuitive interpretation?**
>
> We first note Eq. 5 derives from three key observations:
> 1. Assuming the Q-functions are locally linear, the variance of the Q-values at a direction $\\mathbf{w}$ is minimized when $\\mathbf{w}$ is the eigenvector corresponding to the smallest eigenvalue of the variance matrix.
> 2. In this case, the variance of the Q-values equals the smallest eigenvalue multiplied by $k^2$.
> 3. The smallest eigenvalue is upper-bounded by $\epsilon$ which indicates how well the gradients of the Q-functions are aligned.
>
> These observations result in the relationships below:
> $$
> \\mathrm{Var}(Q\_{\phi_j}(\\mathbf{s}, \\mathbf{a} + k\\mathbf{w})) \ge \\mathrm{Var}(Q_{\\phi_j}(\\mathbf{s}, \\mathbf{a} + k\\mathbf{w}\_{\\mathrm{min}})) = k^2 \\lambda\_{\mathrm{min}}
> $$
>
> $$
> \\mathrm{Var}(Q\_{\\phi_j}(\\mathbf{s}, \\mathbf{a} + k\\mathbf{w}\_{\\mathrm{min}})) = k^2 \\lambda\_{\\mathrm{min}} \le k^2 \\frac{1}{|\\mathcal{A}|} \\frac{N-1}{N} \\epsilon \tag{Eq. 5},
> $$
> where $\\lambda\_{\\mathrm{min}}$ and $\mathbf{w}\_{\\mathrm{min}}$ is the smallest eigenvalue and the corresponding eigenvector of the variance matrix $\\mathrm{Var}(\\nabla\_a Q\_{\phi_j}(\\mathbf{s}, \\mathbf{a}))$. As we only assume the local linearity of the Q-functions, it is correct that Eq. 5 also holds for EDAC. In terms of Eq. 5, EDAC can be interpreted as increasing the RHS of Eq. 5 as it diversifies the gradients (larger $\\epsilon$).
>
> However, we note that increasing this upper bound is not the motivation of EDAC. Our purpose of introducing Eq. 5 was to mathematically show that Q-ensemble networks can fail to sufficiently penalize OOD actions in the neighborhood as the networks’ gradients tend to be well-aligned. On the other hand, the goal of EDAC is to maximize the lower bound for the variance of the Q-values along any direction, which is given by $k^2 \\lambda\_{\mathrm{min}}$ in the first equation above. Since the sum of eigenvalues is less or equal to one by lemma 1 and all the eigenvalues are non-negative, the smallest eigenvalue is always less than $1 / |\\mathcal{A}|$. Therefore, assuming that the smallest eigenvalue is fully maximized by EDAC, we can compute the tight lower bound of the first equation by
> $$
> \\mathrm{Var}(Q\_{\phi_j}(\\mathbf{s}, \\mathbf{a} + k\\mathbf{w})) \ge k^2 \\frac{1}{|\\mathcal{A}|}.
> $$
>
> In practice, we instead maximize the trace to reduce the computational burden. Nevertheless, the modified objective is closely related to maximizing the smallest eigenvalue. As we noted in our paper, maximizing the trace is equivalent to minimizing the cosine similarity of all distinct pairs of the gradients, which makes the gradients uniformly distributed on the unit sphere $S^{|\\mathcal{A}|-1}$. So, the matrix $\\mathrm{Var}(\\nabla\_a Q\_{\\phi\_j}(\\mathbf{s}, \\mathbf{a}))$ can be seen as a sample covariance matrix of a uniform spherical distribution. It can be easily proved that the covariance matrix of a uniform distribution on $S^{n-1}$ is $\\frac{1}{n} I$, whose all eigenvalues are equal to $\\frac{1}{n}$ (please refer to the link below). Therefore, as the number of Q-ensembles goes to infinity, the sample covariance $\\mathrm{Var}(\\nabla\_a Q\_{\\phi_j}(\\mathbf{s}, \\mathbf{a}))$ converges $\\frac{1}{|\\mathcal{A}|} I$, attaining the maximum value for the smallest eigenvalue. We will state this logic concretely in the revised version.
>
> Link: https://stats.stackexchange.com/questions/22764/covariance-matrix-of-uniform-spherical-distribution
>
> **5. Comparison to EMaQ and UWAC.**
>
> Thank you for your suggestion. Compared to the concurrent works mentioned, the core significance of our work is that we show using the Q-ensemble is sufficient to train successful offline RL agents. While EMaQ also utilizes Q-ensemble, the main source of its conservativeness comes from sampling actions from the (estimated) behavior policy. UWAC uses the variance of Q-ensemble networks as an uncertainty measure but still relies on constraining the policy network to lie close to the dataset [1]. In contrast, our method does not apply any additional regularizers apart from the Q-value penalization. We will include these methods as related works and clarify the differences in the revised version.
> Also, we empirically compared our method with UWAC, whose official code is available, and reported the results in Common response 1. We find that our method outperforms UWAC with a large margin on D4RL Gym datasets.
>
> [1] Kumar, et. al., Stabilizing Off-Policy Q-Learning via Bootstrapping Error Reduction, NeurIPS 2019.
>
> **6. The hyperparameter sensitivity of the gradient correction in EDAC.**
>
> To measure the hyperparameter sensitivity of EDAC, we sweep the weight of the gradient diversification term $\\eta$ in the range of $\\{0.0, 0.5, 1.0, 2.0, 5.0\\}$ on the hopper datasets, fixing the number of Q networks to $N=50$, and present the results below.
>
> |  | SAC-$N$  | EDAC | EDAC | EDAC | EDAC |
> |---|---:|---:|---:|---:|---:|
> |  | $\\eta=0$ | $\\eta=0.5$ | $\\eta=1.0$ | $\\eta=2.0$ | $\\eta=5.0$ |
> | random | **25.3±10.4** | 9.3±6.2 | 6.7±0.8 | 3.8±1.5 | 1.9±0.8 |
> | medium | 7.3±0.1 | **102.2±0,4** | **101.6±0.5** | 94.5±12.4 | 75.5±24.1 |
> | expert | 2.3±0.1 | **110.3±0.2** | **110.1±0.1** | **109.8±0.2** | **109.9±0.2** |
> | medium-expert | 46.9±33.0 | **103.8±12.4** | **110.7±0.1** | **109.8±0.2** | **109.8±0.2** |
> | medium-replay | **100.9±0.4** | **100.3±0.8** | **101.0±0.4** | **100.2±0.5** | 20.6±0.7 |
> | full-replay | **105.6±0.4** | **104.9±0.5** | **105.4±0.6** | **104.0±0.2** | **106.3±0.9** |
>
> The results show there exists a large well of hyperparameters where EDAC achieves expert-level performance on all of the datasets except for the random dataset. We also observe that increasing $\\eta$ sometimes degrades the performance on random, medium, and medium-replay datasets which contain trajectories drawn from suboptimal policies. Intuitively, the gradient diversification term induces the learned policy to favor in-distribution actions over OOD actions. Therefore, increasing $\\eta$ can lead to a more conservative policy, which is undesirable if the behavior policy is suboptimal.
>
> **7. Minor comments.**
>
> Thank you for your suggestions. We will revise the paper accordingly in the updated version.

---

> > ### Comment · Reviewer_MGH5 · 2021-08-25
> > **Response to the authors**
> >
> > I appreciate the careful and detailed responses by the authors. While I still have a concern about the relations between the math part and proposed algorithms, I have raised my score respecting extensive empirical evaluations. Also, let me point out a few of my remaining concerns. I hope the authors deal with them.
> >
> > 1. L158 "Note that this assumption can be easily satisfied by optimizing the Bellman error."
> > I don't think this statement is the correct one, since, in practice especially with function approximation, we may observe that the error often accumulates during training. The authors should modify this to a milder tone.
> >
> > 2. I feel it's better to include the discussion on the computational costs the authors provided, in the main text, since it is seemingly costly to take the massive ensemble for SAC-N and EDAC.

---

> > > ### Author Response · Authors · 2021-08-27
> > > **Response #2 to Reviewer MGH5**
> > >
> > > Thank you for the response.
> > >
> > > We are glad to have addressed some of your concerns, and provide our response to your additional suggestions:
> > >
> > > 1. We agree with the reviewer that toning down the current statement would be more appropriate, and will revise our expression in the updated manuscript.
> > >
> > > 2. Thank you for the suggestion. We will include the discussion regarding the computational costs in the main text.
> > >
> > > Please let us know if there are additional questions!

---

### Official Review · Reviewer_jwqx · 2021-07-12

**Rating:** 5
**Confidence:** 4

**Summary:**

This paper considers ensemble actor-critic structure for offline reinforcement learning.
In particular, they focus on SAC-$N$, a version of SAC where the $Q$-function is updated pessimistically w.r.t. a $Q$-function ensemble.
The authors notice that a large value of $N$, SAC-$N$ seems to improve its performance.

This effect is imputed to the fact that the pessimistic estimate is due to the uncertainty expressed by the ensemble (e.g., the higher variance in the ensemble, the more pessimistic is the estimate). Having a large ensemble can result in a higher variance of the estimate, and therefore on a stronger penalization of out of distribution (OOD) state-action pairs.

The authors, therefore, argue that favoring diversity in the ensemble should reach the same effect, without the need of having a large ensemble. To this end, they introduce a regularizer that induces a higher variance of the action-gradients.

The algorithm is validated on different tasks.

**Limitations And Societal Impact:**

The authors discussed the limitations of their proposed approach.
They did not discuss the societal impact, as that does not apply to this work.

**Main Review:**

The literature is rich in papers proposing methods to select "in-distribution" state-action pairs while penalizing ODD state-actions.
The novelties of this paper, as far as I can see, rely on

1) a pessimistic $Q$-estimate using a $Q$-ensemble;
2) the introduction of a new regularizer to induce diversity in the $Q$-ensamble.

While the pessimistic approach in the face of uncertainty still makes sense (but yet explored in the community [1, 2]), there is not a clear logic flow that drives the selection of the regularizer chosen by the authors. One could have selected to favor the variance between the $Q$-estimate for example. The explanation given by the authors is, in my opinion, quite fuzzy.

In short, the contribution of the paper is limited. And the logical argumentation of the algorithm is also weak.

The empirical section is somehow better but still lacks comparison with state-of-the-art algorithms. The algorithm is validated on different, state-of-the-art, benchmarking tasks. The proposed algorithm seems to achieve better results than behavioral cloning, SAC (which is not originally thought for offline policy optimization), REM, CQL.
For such empirical work, I would recommend comparing also with other related algorithms, like BRAC, BCQ, but also MOPO and MOReL.


__Note:__, in __Algorithm 1__ the update rule of the policy (6:) is incorrect. Please, fix it.


[1] Lee, Seunghyun, et al. "Offline-to-Online Reinforcement Learning via Balanced Replay and Pessimistic Q-Ensemble." arXiv preprint arXiv:2107.00591 (2021).
[2] Jin, Ying, Zhuoran Yang, and Zhaoran Wang. "Is Pessimism Provably Efficient for Offline RL?." International Conference on Machine Learning. PMLR, 2021.

UPDATE
--------------------------

I appreciate the effort of the authors in including new baselines and in explaining well their perspective.
I still think that the novelty introduced is limited. However, I must also admit that with the new results, the empirical section will look stronger and I give credit to the author by raising my score accordingly.


**Time Spent Reviewing:**

3.5

---

> ### Author Response · Authors · 2021-08-10
> **Response to Reviewer jwqx**
>
> Thank you for your constructive and thoughtful feedback. We would like to address the reviewer’s concerns below.
>
> **1. The contribution of the paper is limited.**
>
> We respectfully disagree about the limited contribution of our paper. Our paper is the first work in offline RL to demonstrate that considering the Q uncertainty is **sufficient** to train high-performance agents in various benchmarks. While some previous methods also take a pessimistic approach in the face of Q uncertainty, their usage of the uncertainty is limited. These methods still require additional techniques such as CQL-style Q regularization [1] or policy constraints [2, 3]. As noted in L90, the extent to which the Q uncertainty can be exploited has never been explored.
>
> Our work shows that we can outperform the previous methods with complicated constraints by only utilizing the Q uncertainty through a simple penalization technique. This simplicity is a major strength of our method, which allows easy usage in practice. On top of this, we also develop a novel diversification technique, which significantly reduces the required number of Q-networks for the uncertainty penalization. We believe our proposed methods and results will provide valuable insights for future studies and applications.
>
> [1] Lee, et. al., Offline-to-Online Reinforcement Learning via Balanced Replay and Pessimistic Q-Ensemble, arXiv:2107.00591.
>
> [2] Kumar, et. al., Stabilizing Off-Policy Q-Learning via Bootstrapping Error Reduction, NeurIPS 2019.
>
> [3] Wu, et. al., Uncertainty Weighted Actor-Critic for Offline Reinforcement Learning, ICML 2021.
>
> **2. There is not a clear logic flow that drives the selection of the regularizer. One could have selected to favor the variance between the Q-estimate for example.**
>
> We agree with the reviewer that favoring the variance between the Q-value estimates can be a possible option. We had earlier tried to increase the variance of the Q-estimates directly but found out that this regularization sometimes destabilizes the training procedure. The table below shows the average return and the Q-value estimation statistics on the walker2d-expert dataset when using the Q-estimate variance regularizer, compared to our method.
>
> |                       | Return | Avg of Q Estimates | Std of Q Estimates (behavior action) | Std of Q Estimates  (random action) | Q Std Gap |
> |-----------------------|:-------:|:-----------------------:|:-------------------------------------:|:-----------------------------------:|:------------------------------:|
> | Variance reg. ($c$=50)  |    511 |               overflow |                                  N/A |                                N/A |                           N/A |
> | Variance reg. ($c$=100) |     20 |                    -95 |                                  5.3 |                                  7 |                           1.7 |
> | Variance reg. ($c$=200) |    368 |                   -929 |                                 10.6 |                               15.1 |                           4.5 |
> | EDAC                  |   5236 |                    392 |                                  1.2 |                               10.5 |                           9.3 |
>
> **Table 1.** Comparison of EDAC with Q-estimate variance regularization on walker2d-expert dataset. Variance reg. adds to SAC-N a regularizing term that explicitly increases the variance of the Q-estimates, weighted by the coefficient $c$. We use the same number of Q-networks for all methods.
>
> On the walker2d-expert dataset, adding the variance-enhancing regularizer either leads to two results: (1) Exploding Q-values when the regularization is not strong ($c$=50) or (2) severe Q-value underestimation when the regularization is stronger ($c$=100, 200).
> The reason behind these two extreme modes is that the gap of the Q-estimate variance between behavior actions and OOD actions, which is crucial for conservative learning, increases much slower than the absolute increase of the Q-estimate variances. For example, on Variance reg. ($c$=200), the Q-estimate Std gap is 4.5, which is about half of EDAC, whereas the absolute Q-estimate Stds on both actions are much higher. In EDAC, the variance of Q-estimates on behavior actions remains small even though the OOD actions are sufficiently penalized, as we only diversify the Q-networks’ gradients instead of the Q-values themselves. We will explain our logic more concretely in the revised version.
>
> **3. The empirical section is somehow better but still lacks comparison with state-of-the-art algorithms.**
>
> Thank you for the recommendation. We compared our methods with more baselines following your suggestion. Please refer to Common response 1 for the detailed results.
>
> **4. The update rule of the policy (6:) in Algorithm 1 is incorrect.**
>
> Thank you for the pointer, but we could not find where the policy update rule is incorrect. Would you please be kind enough to point which part is wrong in detail?

---

> > ### Comment · Reviewer_jwqx · 2021-08-11
> > **Line 6 Algorithm 1.**
> >
> > I want to thankyou the authors for their clear response. In the following, I want to clarify the last note.
> > I am sorry if I was not clear in the review, I thought that it was obvious. In the update rule that you wrote
> >
> > $$
> >     \nabla_\theta \frac{1}{B} \sum_{b\in B} \min_{j=1\dots n }Q_{\phi_j}(s, a) - \beta \log \pi_\theta(s, a) \quad a \sim \pi_\theta(a|s)
> > $$
> >  which is
> > $$
> > =  - \frac{1}{B} \sum_{b\in B} \min_{j=1\dots n }  \nabla_\theta \beta \log \pi_\theta(s, a) \quad a \sim \pi_\theta(a|s)
> > $$
> > since $Q_\phi$ is considered constant w.r.t. $\theta$. Therefore, the $Q$-function does not even appear in the update.  I am not able to understand what you are trying to do exactly. The policy gradient theorem states that
> >
> > $$
> > \nabla_\theta J(\theta) = \nabla_\theta \mathbb{E}_{\mu_\pi}\left[Q^\pi(s, a) \nabla_\theta \pi_\theta (a|s)\right].
> > $$

---

> > > ### Author Response · Authors · 2021-08-12
> > > **Line 6 Algorithm 1.**
> > >
> > > Thank you for the quick response. Algorithm 1 uses an algorithm format modified from other SAC-based offline RL papers [1, 2], where the purpose of the modification was to help the readers understand the learning process per mini-batch. However, we notice the update rule expression can be adjusted to be more explicit. Our policy update rule can be written in more detail as below:
> > >
> > > $$
> > > \nabla_\theta \frac{1}{\|B\|} \sum_{s \in B} \left( \min_{j=1,\ldots, N} Q\_{\phi_j} (s, \tilde{a}_\theta (s)) - \beta \log \pi_\theta (\tilde{a}_\theta (s) \mid s) \right),
> > > $$
> > >
> > > where $\tilde{a}_\theta (s)$ is a sample from $\pi_\theta (\cdot \mid s)$ which is differentiable w.r.t. $\theta$ via the reparameterization trick.
> > >
> > > We will rewrite the update rule accordingly in the revised version.
> > >
> > > [1] Kumar et al., Conservative Q-Learning for Offline Reinforcement Learning, NeurIPS 2020.
> > >
> > > [2] Kostrikov et al., Offline Reinforcement Learning with Fisher Divergence Critic Regularization, ICML 2021.

---

> > > > ### Comment · Reviewer_jwqx · 2021-08-12
> > > > **Update rule**
> > > >
> > > > Okay, sorry, now I see your point!
> > > >
> > > > Yes, I was a bit confused as I am more used to explicitly state the gradient (using the parametrization trick or the log-likelihood). But now I see your point.
> > > >
> > > > Even though you correctly report the formula being used in previous work, I still think that it is not clear. In the previous notation, one could think that you first sample and then taking the gradient.
> > > > I personally think that the best way to make sure that the gradient is also considering the distribution is something like
> > > >
> > > > $$ \nabla_\theta \frac{1}{|B|} \sum_{s \sim B; a \sim \pi_{\theta}} Q(s, a) -\beta\log\pi_\theta(a|s).$$
> > > >
> > > > Thank you very much for taking the time to consider my concern. I know I have been bothering you with a small issue, but the equation looked just wrong to me.

---

> > > > > ### Author Response · Authors · 2021-08-27
> > > > > **Response #3 to reviewer jwqx**
> > > > >
> > > > > Thank you for the response.
> > > > >
> > > > > Your feedback has helped us improve our paper in terms of conciseness. We also appreciate your suggesting a notation for the update rule of our algorithm.
> > > > >
> > > > > Please let us know if there are additional questions!

---

### Official Review · Reviewer_PrWt · 2021-07-16

**Rating:** 7
**Confidence:** 4

**Summary:**

In this work the authors propose a new algorithm for the problem of offline RL. One of the main
difficulties in offline RL is estimating the Q-values of Out-Of-Distribution action state pairs.
To overcome this, they authors propose a new actor critic method, where an ensemble of Q values for the critic
is trained  using clipped Q-learning. They show that this can be used to penalize OOD data
with high prediction uncertainties. The resulting algorithm achieves state-of-the-art performance in
a range of D4RL benchmarks. Furthermore, in order to reduce the size of the ensemble required to
achieve good performance they propose a modification on the critic loss. They show that the resulting
algorithm called Ensemble-Diversified Actor-Critic can match the performance of the naive approach while
using only one tenth of the number of ensemble Q values.

**Limitations And Societal Impact:**

Yes

**Main Review:**

The authors propose a simple but novel approach to the problem of offline RL, while reporting suprisingly
good results. The use of ensembles in offline RL is a interesting approach and opens up new research avenues.

Some comments on the paper:

In figure 1, increasing the number of ensembles hurts performance in halfcheetah-medium but helps in hopper-medium.
The authors shoudld provide an explanation for this result.

line 84: Mentions the sampling of the action distribution as computationally expensive which can be a limiting factor.
However, maintaining multiple Q values and calling inference on them can incurr an even higher cost, so this cannot be
used as an argument in favor of their method.

line: 163: Define Proposition 1 before using it.

**Time Spent Reviewing:**

3

---

> ### Author Response · Authors · 2021-08-10
> **Response to Reviewer PrWt**
>
> Thank you for your encouraging comments and constructive feedback. We would like to address your comments below.
>
> **1. Figure 1., Explanations on why increasing the number of ensembles hurts performance in halfcheetah-medium but helps in hopper-medium.**
>
> Increasing the number of ensembles implies enforcing our agent to be more pessimistic. While a certain level of pessimism is crucial for offline RL algorithms, too much pessimism can be detrimental since the agent will stay too close to the dataset. However, we note that the performance degradation is not severe within a reasonable range of $N$, as shown in the halfcheetah-medium plot of Figure 1.
>
> **2. Line 84: Maintaining multiple Q values and calling inference on them can incur an even higher cost, so the sampling of the action distribution cannot be used as an argument in favor of their method.**
>
> Thank you for the suggestion. We compared the computational cost of our methods with CQL and provided the results in Common response 2. In short, our method EDAC runs faster than CQL, with comparable memory consumption. This result is because the inference to the Q-networks is fully parallelizable, which minimizes the increase of runtime with the number of Q-networks. We will include this analysis in our revised manuscript and clarify our argument.
>
> **3. Line 163: Define Proposition 1 before using it.**
>
> Thank you for pointing this out. We will adjust the content order in the revised version.

---

### Official Review · Reviewer_DiJ9 · 2021-07-17

**Rating:** 8
**Confidence:** 4

**Summary:**

The paper presents a simple but rather effective idea (probably known by many but never formally tried): having an ensemble of N Q-networks and then using their *minimum* in the Bellman backup. This is roughly analogous to penalizing the objective function of fitted-Q iteration by the variance of the ensemble, which causes a somehow lower-confidence bound on the learning, which in turn causes conservative values for out-of-distribution actions at each seen state. This framework is then used as part of a soft actor-critic algorithm, called SAC-N. This significantly outperforms SOTA (but much slower in convergence rate). However, this setup can require an extremely large ensemble, which can be a limitation in practice. The authors next observe that the performance of SAC-N is highly correlated with the diversity of the Q-functions’ gradient w.r.t. actions (increased by the size of ensemble, N). To maximize diversity, they proposed to minimize the total cosine similarity matric, which gives rise to maximizing the variance for smallest eigen value of the variance-gradient matrix. As it gets into a computationally intractable problem, they instead maximize the sum of all eigenvalues, which is equal to the total variance. This final matric is called Ensemble Similarity and the resulting algorithm is named Ensemble-Diversified Actor Critic (EDAC). Finally, several experimental results are presented, largely outperforming previous methods.

**Limitations And Societal Impact:**

The paper is mostly technical, so no direct societal impact is expected at this stage. However, when a method like the one proposed in this submission is deployed in practice, it can cause significant impact in high-risk domains like healthcare (is specially so because offline RL largely targets healthcare). Perhaps it could be helpful to have a discussion on how to use the variance as a measure of uncertainty to warn against situations where the learned value/policy is not trustworthy.

**Main Review:**

The paper presents several interesting ideas and nice results. I quite enjoyed reading the manuscript. Here are some comments to help improving the paper:

-- An observation from Figure 2 is that the *deviation from minimum* increases for OOD actions as training advances. This implies that the average value is diverging for those actions; i.e., the overestimation of their values grow with more training. It would be interesting to perhaps suppress this as well, for example by a weak regularization proportional to the size of Q itself. In a related note, this problem could also be due to value overflow. The following paper has a discussion on value overflow in deep RL (which is different from value overestimation):

> Fatemi, M., Sharma, S., van Seijen, H. & Kahu, S. E. (2019), Dead-ends and Secure Exploration in Reinforcement Learning, In International Conference on Machine Learning (pp. 1873-1881).

-- The narrative of section 3 (specially the last part) is rather descriptive than formal. This is fine as long as it gets noted that most of the results are indeed conjectures based upon the experimental results. While I personally enjoyed reading this section too, the claims should be tailored in a correct way. For example, L134: *… we confirm that …* -> is too strong; nothing has been confirmed, but just a nice trend has been observed (which seems to be correct, yet not formally proved).

-- Figure 3: there is a sharp break point for average return at N=100. Is there any explanation for this dramatic change of slope? In particular, both before and after this point, the trend looks surprisingly linear.



**Time Spent Reviewing:**

8

---

> ### Author Response · Authors · 2021-08-10
> **Response to Reviewer DiJ9**
>
> Thank you for your encouraging comments and constructive feedback. We would like to address your comments below.
>
> **1. The deviation from minimum increases for OOD actions as training advances. It would be interesting to perhaps suppress this as well.**
>
> Thank you for the suggestion. Suppressing the average Q-values for OOD actions is indeed a valuable direction for future research. It will also be possible to consider combining our methods with techniques to mitigate the value overflow problem of Q-learning proposed in [1].
>
> [1] Fatemi, et. al., Dead-ends and Secure Exploration in Reinforcement Learning, ICML 2019.
>
>
> **2. The claims in section 3 should be tailored in a correct way.**
>
> Thank you for the suggestion. We will revise our claims and expressions accordingly.
>
> **3. Figure 3: There is a sharp break point for average return at $N$=100. Is there any explanation for this dramatic change of slope?**
>
> We observe that for $N\leq$100, the penalization effect of the clipped Q-learning is not strong enough to prevent the Q-values from exploding. However, as we increase $N$ to more than 100, the Q-values stop overshooting with the enhanced penalization. This difference (whether the Q-values explode or not) is the main reason for the sharp breakpoint on the average return plot. We also note that the x-axis of Figure 3 is in log-scale, so the relationship is not strictly linear. We will add these explanations in the revised version.
>
> **4. It could be helpful to have a discussion on how to use the variance as a measure of uncertainty to warn against situations where the learned value/policy is not trustworthy.**
>
> Thank you for the suggestion. In risk-critical domains such as healthcare, warning the agent ahead of risky situations is of great importance, and we believe our uncertainty-based approach will benefit these cases. For example, agents can make uncertainty-aware decisions based on the variance of Q-values. We will add related discussions in the revised version.

---

### Author Response · Authors · 2021-08-10
**Dear reviewers**

We thank the reviewers for their encouraging and thoughtful comments. We would first like to address some of the common concerns raised by multiple reviewers and address other specific concerns on individual responses.

## Common response

**1. More baselines**

We additionally compared our methods with more baselines on D4RL Gym datasets. First, we add comparisons with some of the well-known offline RL methods, BCQ, BEAR, BRAC, and MOReL. Also, we include the results of UWAC [1], a concurrent work that also utilizes Q-value uncertainty. We reproduced all the methods by following the hyperparameter search procedure listed in each paper and selected the best results. We report the normalized average return results below.

|                           |       BCQ |      BEAR |      BRAC |     MOReL |      UWAC |       CQL |     SAC-$N$ |      EDAC |
|---------------------------|----------:|----------:|----------:|----------:|----------:|----------:|----------:|----------:|
| halfcheetah-random        |   2.2±0.0 |  12.6±1.0 |  24.3±0.7 |  **38.9±1.8** |   2.3±0.0 |  31.3±3.5 |  28.0±0.9 |  28.4±1.0 |
| halfcheetah-medium        |  46.6±0.4 |  42.8±0.1 |  51.9±0.3 |  60.7±4.4 |  43.7±0.4 |  46.9±0.4 |  **67.5±1.2** |  **65.9±0.6** |
| halfcheetah-expert        |  89.9±9.6 |  92.6±0.6 | 39.0±13.8 |  8.4±11.8 |  94.7±1.1 |  97.3±1.1 | **105.2±2.6** | **106.8±3.4** |
| halfcheetah-medium-expert |  95.4±2.0 |  45.7±4.2 |  52.3±0.1 | 80.4±11.7 |  47.0±6.0 |  95.0±1.4 | **107.1±2.0** | **106.3±1.9** |
| halfcheetah-medium-replay |  42.2±0.9 |  39.4±0.8 |  48.6±0.4 |  44.5±5.6 |  38.9±1.1 |  45.3±0.3 |  **63.9±0.8** |  **61.3±1.9** |
| halfcheetah-full-replay   |  69.5±4.0 |  60.1±3.2 |  78.0±0.7 |  70.1±5.1 |  65.1±0.5 |  76.9±0.9 |  **84.5±1.2** |  **84.6±0.9** |
| hopper-random             |   7.8±0.6 |   3.6±3.6 |   8.1±0.6 | **38.1±10.1** |   2.6±0.3 |   5.3±0.6 |  31.3±0.0 | 25.3±10.4 |
| hopper-medium             |  59.4±8.3 |  55.3±3.2 |  77.8±6.1 | 84.0±17.0 |  52.6±4.0 |  61.9±6.4 | **100.3±0.3** | **101.6±0.6** |
| hopper-expert             |   109±4.0 | 39.4±20.5 | 78.1±52.3 | 80.4±34.9 | **111.0±0.8** | 106.5±9.1 | 110.3±0.3 | 110.1±0.1 |
| hopper-medium-expert      | 106.9±5.0 |  66.2±8.5 |  81.3±8.0 | 105.6±8.2 |  54.8±3.2 | 96.9±15.1 | **110.1±0.3** | **110.7±0.1** |
| hopper-medium-replay         | 60.9±14.7 | 57.7±16.5 | 62.7±30.4 | 81.8±17.0 | 31.1±14.8 |  86.3±7.3 | **101.8±0.5** | **101.0±0.5** |
| hopper-full-replay        | 46.6±13.0 | 54.0±24.0 | **107.4±0.5** | 94.4±20.5 |  21.9±8.4 | 101.9±0.6 | 102.9±0.3 | 105.4±0.7 |
| walker2d-random           |   4.9±0.1 |   4.3±1.2 |   1.3±1.4 |  16.0±7.7 |   1.5±0.3 |   5.4±1.7 |  **21.7±0.0** |  **16.6±7.0** |
| walker2d-medium           |  71.8±7.2 | 59.8±40.0 | 59.7±39.9 | 72.8±11.9 |  66.0±9.0 |  79.5±3.2 |  **87.9±0.2** |  **92.5±0.8** |
| walker2d-expert           | 106.3±5.0 | 110.1±0.6 | 55.2±62.2 | 62.6±29.9 | 108.4±0.5 | 109.3±0.1 | 107.4±2.4 | **115.1±1.9** |
| walker2d-medium-expert    | 107.7±3.8 | 107.0±2.9 |  9.3±18.9 | 107.5±5.6 | 85.7±14.0 | 109.1±0.2 | **116.7±0.4** | **114.7±0.9** |
| walker2d-medium-replay    |  57.0±9.6 |  12.2±4.7 | 40.1±47.9 | 40.8±20.4 |  27.1±9.6 | 76.8±10.0 |  **78.7±0.7** |  **87.1±2.3** |
| walker2d-full-replay      | 71.0±21.8 | 79.6±15.6 |  96.9±2.2 | 84.8±13.1 | 60.7±15.6 |  94.2±1.9 |  94.6±0.5 |  **99.8±0.7** |
| Average                   |      64.2 |      52.4 |        54 |      65.1 |      50.8 |      73.7 |      **84.5** |      **85.2** |

The results show our methods outperform all the baseline methods on most of the datasets considered. Also, we reiterate that while the performance of EDAC is marginally better than SAC-$N$, EDAC achieves this result with a much smaller Q-ensemble size.

[1] Wu, et. al., Uncertainty Weighted Actor-Critic for Offline Reinforcement Learning, ICML 2021.

**2. Computational cost of the proposed algorithm**

We compared the computational cost of our methods with CQL on hopper-medium-v2, where our methods require the largest number of Q-networks. For each method, we measure the runtime per epoch along with GPU memory consumption. We run our experiments on a single machine with one RTX3090 GPU and provide the results below.

|         | Runtime (s/epoch) | GPU Memory Consumption (GB) |
|---------|:-----------------:|:---------------------------:|
| SAC     |        21.4       |             1.3             |
| CQL     |        38.2       |             1.4             |
| SAC-500 |        44.1       |             5.1             |
| EDAC    |        30.8       |             1.8             |

As the result shows, our method EDAC runs faster than CQL with comparable memory consumption. Note that CQL is about twice as slower than vanilla SAC due to the additional computations for Q-value regularization (e.g., dual update and approximate logsumexp via sampling). Meanwhile, the inference to the Q-network ensemble in SAC-$N$ and EDAC is embarrassingly parallelizable, minimizing the runtime increase with the number of Q-networks. Also, we note that our gradient diversification term in L185 has linear computational complexity, as we can reformulate the term using the sum of the gradients.

---

### Decision · Program_Chairs · 2021-09-27

**Decision:**

Accept (Poster)

**Comment:**

The authors note that using an ensemble with N-networks in SAC and using the minimum for the Bellman backup performs well on Offline RL. This is a simple approach and interesting observation. As using a large N is computationally expensive, they develop an approach to achieve similar results with many fewer networks.

After the author response, all reviewers and I found the empirical section to clearly demonstrate the strength of their approach. Although there are some concerns on novelty, I think the strong empirical results and simplicity of their approach makes this paper a worthwhile contribution. The clarifications made during the response phase were important, so I encourage the authors to revise the paper with these in mind.